# Ecological control of nitrite in the upper ocean

Emily J. Zakem [1,2], Alia Al-Haj [3], Matthew J. Church[4], Gert L. van Dijken [5], Stephanie Dutkiewicz[1], Sarah Q. Foster [3], Robinson W. Fulweiler [3,6], Matthew M. Mills [5] & Michael J. Follows[1]

Microorganisms oxidize organic nitrogen to nitrate in a series of steps. Nitrite, an intermediate product, accumulates at the base of the sunlit layer in the subtropical ocean, forming a primary nitrite maximum, but can accumulate throughout the sunlit layer at higher latitudes. We model nitrifying chemoautotrophs in a marine ecosystem and demonstrate that microbial community interactions can explain the nitrite distributions. Our theoretical framework proposes that nitrite can accumulate to a higher concentration than ammonium because of differences in underlying redox chemistry and cell size between ammonia- and nitrite-oxidizing chemoautotrophs. Using ocean circulation models, we demonstrate that nitrifying microorganisms are excluded in the sunlit layer when phytoplankton are nitrogen-limited, but thrive at depth when phytoplankton become light-limited, resulting in nitrite accumulation there. However, nitrifying microorganisms may coexist in the sunlit layer when phytoplankton are iron- or light-limited (often in higher latitudes). These results improve understanding of the controls on nitrification, and provide a framework for representing chemoautotrophs and their biogeochemical effects in ocean models.

[1] Department of Earth, Atmospheric and Planetary Sciences, Massachusetts Institute of Technology, Cambridge, MA 02139, USA. [2] Department of Biological Sciences, University of Southern California, Los Angeles, CA 90089, USA. [3] Department of Earth and Environment, Boston University, Boston, MA 02215, USA. [4] Flathead Lake Biological Station, University of Montana, Polson, MT 59860, USA. [5] Department of Earth System Science, Stanford University, Stanford, CA 94305, USA. [6] Department of Biology, Boston University, Boston, MA 02215, USA. Correspondence and requests for materials should be addressed to E.J.Z. (email: ezakem@mit.edu)

Nitrogen proximally limits primary production in much of the surface ocean, and the nitrogen cycle exerts a strong influence on the coupled cycles of carbon and other elements[1]. Most fixed nitrogen in the ocean is in the oxidized form of nitrate ($NO_3^-$). Primary producers and other microorganisms in the surface ocean consume and reduce $NO_3^-$ to build organic molecules. Microorganisms then oxidize detrital organic nitrogen back to $NO_3^-$ in a series of steps, with intermediates ammonium ($NH_4^+$, here considered interchangeable with ammonia, $NH_3$) and nitrite ($NO_2^-$)[2].

Despite its typically low concentration, $NO_2^-$ plays a central role in global nitrogen and carbon cycles, providing a key resource for significant microbial metabolisms. $NO_2^-$ is an intermediate of nitrification, the two-step, microbially mediated oxidation of $NH_4^+$ to $NO_3^-$ ($NH_4^+ \rightarrow NO_2^- \rightarrow NO_3^-$) that occurs in association with distinct clades of metabolically diverse chemoautotrophic archaea and bacteria in many environments[2–5].

The primary nitrite maximum (PNM), an accumulation of $NO_2^-$ at the base of the euphotic (sunlit) zone at concentrations of 10–1000 nmol $L^{-1}$, is ubiquitous in oxygenated subtropical oceans[2] (Fig. 1). In a typical subtropical vertical profile, the PNM is located at the onset of the nitricline, just below the deep chlorophyll maximum (DCM) and an associated $NH_4^+$ maximum[6–11]. The relative magnitude of the $NH_4^+$ maximum varies, but peak $[NH_4^+]$ is consistently lower than $[NO_2^+]$ at the PNM in oligotrophic environments[10,12,13]. These subsurface maxima form in strongly stratified water columns, i.e., when the euphotic zone is deeper than the mixed layer[14,15]. Nitrification rates and the biomass of nitrifying microbes are often observed to peak at, or just below, the PNM[10,13,16–22]. In contrast, in subpolar regions, $NO_2^-$ concentrations are elevated to similar magnitudes throughout all of the upper ocean, including the surface (Fig. 1). In low-oxygen environments, a secondary $NO_2^-$ maximum forms below the PNM at much higher concentrations (order 10 μmol $L^{-1}$) due to anaerobic activity[2]; however, we focus here only on the dynamics of aerobic environments.

Despite the widespread occurrence of the PNM, the mechanisms that determine the locations and magnitude of accumulated $NO_2^-$ are still not fully resolved. What is the source of the accumulated $NO_2^-$ in oxygenated waters? Two hypotheses, not mutually exclusive, have been advanced[23]: (i) excretion of $NO_2^-$ due to incomplete assimilatory reduction of $NO_3^-$ by phytoplankton[14,24,25], and (ii) chemoautotrophic $NH_4^+$ oxidation, the first step of nitrification[23,26]. Here, we focus on outstanding questions of the latter hypothesis, since isotopic evidence suggests that $NH_4^+$ oxidation is a major source of $NO_2^-$ at the PNM[10,21].

First, why does peak nitrification occur at the base of the euphotic zone? Photoinhibition of nitrifying microorganisms has been documented[26–30] and incorporated into ecosystem models[31,32], but observations of nitrification in the euphotic zone[16,18,33] and close to the sea surface[34–36] suggest that this is not universally the case. An alternate (but not exclusive) hypothesis is that phytoplankton outcompete slow-growing, chemoautotrophic nitrifiers for $NH_4^+$ (and $NO_2^-$) in the euphotic zone[37,38], but not deeper where light limits photoautotrophy.

Second, while both $NO_2^-$ and $NH_4^+$ can accumulate in the oxygenated thermocline, why does $NO_2^-$ consistently accumulate to a higher concentration in oligotrophic environments? Hypotheses include differential effects of photoinhibition of $NH_4^+$ and $NO_2^-$ oxidation[26,39,40] and differential temperature sensitivity of their rates[41], though some observations conflict with the latter[42]. Here, we will suggest that this reflects differences in the metabolisms of the distinct clades of nitrifiying microorganisms.

Third, why does nitrification sometimes occur in the euphotic zone, and why is surface $[NO_2^-]$ also elevated in some areas, such as in subpolar regimes? Possible explanations include accumulation due to rapid entrainment of $NO_2^-$ from below the euphotic zone, chemoautotrophic nitrification within the euphotic zone, or the incomplete reduction of $NO_3^-$ by phytoplankton.

Here we synthesize and address these questions and hypotheses using a hierarchy of mathematical models and simulations that are grounded in theoretical and laboratory evaluations of the kinetics and efficiencies of marine nitrifying microorganisms. The models are used to interpret both existing and new water column observations in the subtropical North Pacific as well as the global distribution of $NO_2^-$ in the oxygenated upper ocean.

We first present a general population dynamics model for $NH_4^+$ and $NO_2^-$ oxidizers. The model couples estimates of cellular substrate uptake rates with an energetically informed stoichiometry of whole-organism metabolism. We then examine point balance solutions of the model and find that the relative concentrations of $NH_4^+$ and $NO_2^-$ near the PNM reflect the respective subsistence concentrations for $NH_4^+$-oxidizing and $NO_2^-$-oxidizing organisms, with exact concentrations also influenced by vertical mixing. We next discuss how competitive interactions control the vertical structure of nitrification and the position of the subtropical PNM using simulations of water column profiles of an upper ocean microbial ecosystem. Finally, we implement the ecosystem model in a global ocean simulation, and find that deep mixed layers and the light or iron limitation of primary producers can explain $[NO_2^-]$ and nitrification at the surface in some regions.

## Results

**Microbial population dynamics.** We consider the clades of microorganisms carrying out the two steps of nitrification as two metabolic functional types: ammonia-oxidizing and nitrite-oxidizing organisms (hereafter, AOO and NOO, respectively). We describe the population dynamics of each type as:

$$\frac{\partial B_i}{\partial t} = \underbrace{\mu_i(R)B_i}_{\text{Growth}} - \underbrace{L_i(Z)B_i}_{\text{Loss}} - \underbrace{\nabla \cdot (\mathbf{u}B_i) + \nabla \cdot (\mathbf{K}\nabla B_i)}_{\text{Advection and mixing}} \quad (1)$$

where $B_i$ (mol $L^{-1}$) is the biomass of type $i$ with population growth rate $\mu_i$ ($d^{-1}$), loss rate $L_i$ ($d^{-1}$), and physical transport. Loss rate is a function of the population density of grazers $Z$ and other factors (Methods). We describe population growth rate $\mu_i$ using Monod kinetics with limiting resource $R_j$ (determined by Liebig's Law of the minimum). Growth rate depends on the yield of biomass with respect to the limiting resource $y_{ij}$ (mol biomass mol $R^{-1}$), the maximum specific resource uptake rate $V_{\max_{ij}}$ (mol $R$ per mol biomass per day), and a half-saturation concentration $K_{ij}$ (mol $L^{-1}$) as:

$$\mu_i = y_{ij} V_{\max_{ij}} \frac{R_j}{R_j + K_{ij}}. \quad (2)$$

The uptake parameters together give an expression for the specific uptake affinity ($V_{\max_{ij}} K_{ij}^{-1}$), a measure of competitive strength at low resource concentrations[43]. For the two nitrifying metabolisms, we assume that DIN and oxygen are the potentially limiting resources ($R_j$). Over most of the subtropical thermocline, oxygen exists at sufficiently high concentrations (greater than nanomolar) to serve as the terminal electron acceptor for AOO and NOO[44,45].

Equations (1) and (2) provide a general description of microbial population dynamics. We next consider idealized approximations and numerical solutions to interpret the controls on $NH_4^+$ and $NO_2^-$ in the subsurface ocean.

**Point balance solution for $[NH_4^+]:[NO_2^-]$ at the PNM.** Why does $NO_2^-$ often accumulate to a higher degree than $NH_4^+$ at the

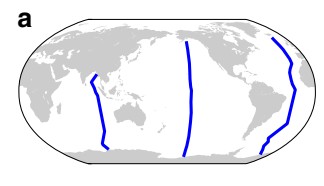

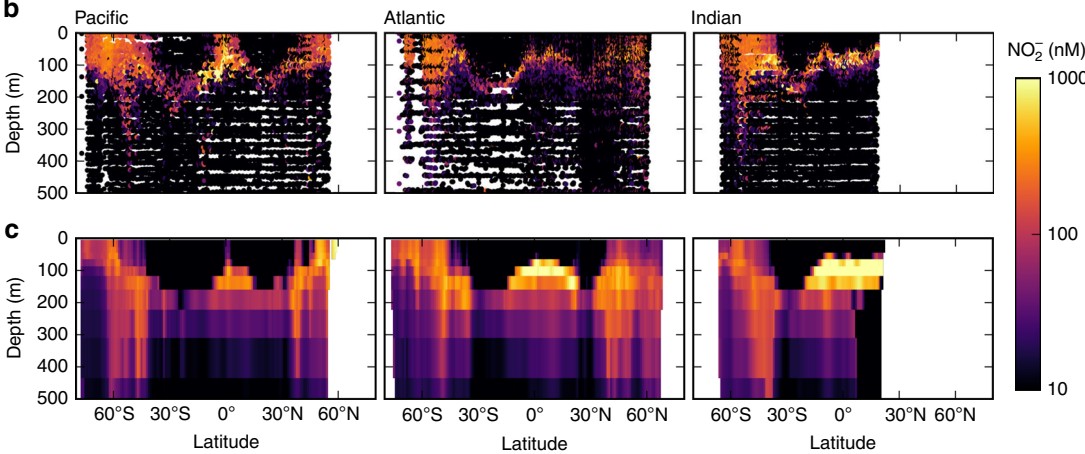

**Fig. 1** Observed and modeled nitrite ($NO_2^-$) concentrations along three transects in the ocean. **a** Map of the GLODAPv2 transect locations. **b** Transects from the GLODAPv2 database[60, 61]. **c** Transects from the global simulation. Map generated with Python version 3.5.1, Matplotlib version 1.5.1, and Basemap version 1.0.7[86]

PNM[10,12,13]? Consider the local dynamics of the subsurface oligotrophic environment, assuming the simplest approximation of the model, in which physical transport is negligible and the system is close to steady state ($\partial B_i/\partial t \sim 0$). Using Eqs. (1) and (2), we evaluate the concentration of resource $R_j$ that limits the growth of nitrifier type $i$ as:

$$R_{ij}^* = \frac{K_{ij}L_i}{y_{ij}V_{\max_{ij}} - L_i}. \quad (3)$$

$R_{ij}^*$ is the subsistence concentration of type $i$[46,47]. In theory, at a steady state, the population with the lowest $R^*$ excludes all others limited by the same resource (if its maximum growth rate, $y_{ij}V_{\max_{ij}}$, is larger than loss rate $L_i$), and the environmental concentration is set to this $R_{ij}^*$. We hypothesize that in the vicinity of the PNM, $NH_4^+$ and $NO_2^-$ are the respective limiting resources for AOO and NOO, respectively, and that the ratio of the environmental concentrations will therefore reflect the ratio of their respective subsistence concentrations.

What are the differences in $R_{ij}^*$ for the nitrifier guilds? We first estimate the yields of the AOO and NOO functional types. Following established methodology[48], we derive stoichiometrically balanced equations to describe catabolic and anabolic processes as a function of the redox chemistry that underlies each step of nitrification. Constraining thermodynamic efficiency with published laboratory observations of marine AOO and NOO growth, we approximate the population-level growth stoichiometries of ammonia-oxidizing biomass $B_{AOO}$ and nitrite-oxidizing biomass $B_{NOO}$ as:

$$(112 \pm 22)NH_4^+ + (162 \pm 32)O_2 \rightarrow B_{AOO} + (111 \pm 22)NO_2^-, \quad (4)$$

$$(334 \pm 67)NO_2^- + NH_4^+ + (162 \pm 32)O_2 \rightarrow B_{NOO} + (334 \pm 67)NO_3^-, \quad (5)$$

where biomasses are assumed to be synthesized from one mole of $NH_4^+$ for both types[4] (Methods). The predicted stoichiometry of $NH_4^+$ and $O_2$ demand for the AOO is 1:1.45, close to that observed in culture[37] (1:1.52) indicating that the model captures key aspects of real metabolisms. In Eqs. (4) and (5), the yields for AOO and NOO are $y_{AOO,NH_4} = 112^{-1}$ and $y_{NOO,NO_2} = 334^{-1}$ moles biomass N synthesized per mole DIN used, respectively. In other words, the NOO type consumes three times as much $NO_2^-$ to produce a unit of biomass, relative to the consumption of $NH_4^+$ by the AOO type.

Second, we speculate that NOO specific uptake affinity may be lower than that of AOO if (as is observed) $NO_2^-$-oxidizing bacteria are larger than $NH_4^+$-oxidizing archaea[37,49,50]. Established empirical and theoretical allometric relationships[51,52] suggest that specific affinity decreases with increasing cell radius $r$ as $r^{-2}$ (Methods). If we assume a 10-fold larger cell volume for NOO, this estimates a 4.6-fold lower specific affinity.

Using these differences in yield ($y_{ij}$) and affinity ($V_{\max_{ij}}K_{ij}^{-1}$), we evaluate the ratio of the subsistence concentrations of $NH_4^+$ and $NO_2^-$ for the AOO and NOO, respectively, in Eq. (3). If AOO and NOO are of similar size/affinity, and if loss rates ($L_i$) are low relative to maximum growth rates (and thus negligible in the denominator), then we predict that $[NH_4^+]:[NO_2^-] \sim y_{AOO,NH_4} : y_{NOO,NO_2} \sim 1:3$ in the vicinity of the PNM. This is consistent with the observed ratio at the PNM in oligotrophic environments[10,12]. If yields are similar, but affinity is 4.6-fold different, we predict a similar order of enhanced $NO_2^-$ accumulation. Together, differences in both yield and affinity would result in an even lower ratio, underestimating $[NH_4^+]:[NO_2^-]$. Below, we examine the independent and additive effects of both distinctions between the nitrifier metabolisms in the water column model.

**Understanding vertical structure with a water column model.** What sets the vertical structure of the PNM? To address this, we simulate an idealized oligotrophic water column in one dimension, in which the photon flux attenuates with depth and vertical

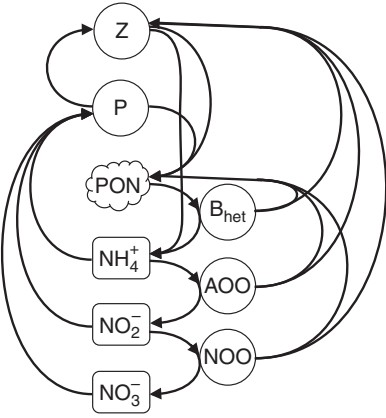

**Fig. 2** Schematic of the marine ecosystem model with explicit nitrification. The model resolves three species of inorganic fixed nitrogen (ammonium ($NH_4^+$), nitrite ($NO_2^-$), and nitrate ($NO_3^-$)), particulate organic nitrogen (PON), phytoplankton (P), zooplankton (Z), heterotrophic bacteria ($B_{het}$), ammonia-oxidizing organisms (AOO), and nitrite-oxidizing organisms (NOO)

mixing of biomass and resources is described by turbulent diffusion. Enhanced vertical mixing close to the surface simulates the mixed layer.

Our ecosystem model resolves the interactions and mixing of inorganic and organic nitrogen and nitrogen-based biomass of five microbial metabolic functional type populations (Fig. 2). AOO and NOO are modeled as above (parameter values in Supplementary Table 1), with growth rate limited by the supply of $NH_4^+$ or $NO_2^-$, the supply of oxygen[45], or internal constraints (i.e., the maximum rate). For simplicity, we neglect other limitations such as iron availability (Supplementary Note 3). A phytoplankton functional type assimilates $NH_4^+$, $NO_2^-$, and $NO_3^-$ into biomass with light- and nutrient-limited growth[53,54]. The traits of the phytoplankton type are based on the cyanobacterium *Prochlorococcus*, since it is often the most abundant photoautotroph in oligotrophic marine environments, and perhaps the strongest photoautotrophic competitor for DIN due to its high-nutrient affinity, a consequence of its small size[52,55]. A heterotrophic bacterial type remineralizes organic detritus to $NH_4^+$. A microzooplankton grazer type consumes the other microbes and excretes $NH_4^+$. Temperature modifies the rates of all organisms except for the nitrifying types, following experimental results[56] (Supplementary Note 1). A pool of sinking organic detritus is produced from the mortality of all populations. We do not impose photoinhibition or any other direct inhibitor of nitrification, and instead examine the vertical structure that emerges as a function of the ecological interactions.

For comparison with the model, we show relevant observations collected at several stations in the North Pacific subtropical gyre in 2014 (Fig. 3; station locations in Supplementary Fig. 1). Peak [$NO_2^-$], detectable at two out of four stations, is positioned below the DCM, reaching 110 and 130 nmol L$^{-1}$. Detectable $NH_4^+$ oxidation rates peak at this apparent PNM at these locations, reaching 3 and 6 nmol L$^{-1}$ d$^{-1}$. Gene abundances indicating $NH_4^+$-oxidizing archaea (*amoA*) also correlate with [$NO_2^-$]. In general, the resulting vertical structures are consistent with other observations of the PNM in oligotrophic systems[10,12].

**The emergent PNM**. With the subtropical configuration, a PNM emerges consistently in the water column as a consequence of the

ecological interactions. We illustrate the model equilibrium solution that corresponds to our observed profile from the North Pacific (Fig. 3; full solutions in Supplementary Fig. 2). The simulation qualitatively reproduces the vertical structures of [$NO_2^-$], rates of $NH_4^+$ oxidation, and AOO abundances. We explored the sensitivity of the model solution with an ensemble of ~1000 model realizations in which ecosystem model parameters were sampled randomly over reasonable ranges of uncertainty (see Methods and Supplementary Table 1). The s.d. of the ensemble of solutions (Fig. 3, shaded region) shows that the key results outlined below are indeed qualitatively robust despite the uncertainties.

With the steady-state solutions, we calculate the subsistence concentrations ($R^*$s) to diagnose ecological control (Fig. 4). We use Eq. (3) for AOO and NOO, with maximum growth rate $y_{ij}V_{max_{ij}}$, and Eq. (28) for phytoplankton, incorporating their maximum light-limited growth rate (Methods). We then use the resulting loss rate $L$, a function of the dynamic grazer population $Z$ (Eq. (29)), to calculate the depth-varying $R^*$s for $NH_4^+$ and $NO_2^-$.

We find that the modeled phytoplankton exclude the chemoautotrophic nitrifiers in the euphotic zone, where both are limited by nitrogen. This is because the photoautotrophs have lower subsistence concentrations for $NH_4^+$ and $NO_2^-$ at the surface (Fig. 4). Theory and observations show that $NH_4^+$-oxidizing archaea and picophytoplankton have similar uptake affinities for $NH_4^+$[37,51,52]. However, for picophytoplankton, the effective half-saturation concentrations for nitrogen uptake with respect to growth rate are significantly lower[57], and maximum growth rate is higher. This reflects that nitrifiers use DIN for energy production, which is relatively inefficient, while the phytoplankton use it as a nitrogen source for synthesis with a much higher biomass yield.

Deeper in the water column, once light limits photoautotrophy, the maximum light-limited growth rate decreases significantly, leading to sharply increasing $R^*$s (the green dashed lines in Fig. 4). With still increasing depth, phytoplankton losses become larger than their growth rate (Supplementary Fig. 3c), and the phytoplankton can no longer survive (where the green dashed lines end in Fig. 4). Once phytoplankton are excluded, the nitrifiers become competitive for $NH_4^+$ and $NO_2^-$, and the ambient nutrient concentrations are set by the nitrifier $R^*$s instead. This accounts for the large increase in [$NH_4^+$] and [$NO_2^-$] at about 100 m (in this example).

$R^*$s decrease with depth as grazing pressure is reduced (see blue dashed line in Supplementary Fig. 3c), and so, after the increase, $NH_4^+$ and $NO_2^-$ then decline with depth. Thus, subsurface maxima in both $NO_2^-$ (the PNM) and $NH_4^+$ emerge in the simulations, controlled by a combination of top–down and bottom–up processes. As in the point balance, $NO_2^-$ accumulates to a higher maximum concentration than $NH_4^+$ because of the yield and affinity distinctions between AOO and NOO.

We further demonstrate the control of [$NH_4^+$]:[$NO_2^-$], and its sensitivity to the nitrifier parameters, using model experiments in which we isolate the differences in yields and affinities of the NOO and AOO (Fig. 3e–g). With both the yield and affinity differences between the nitrifiers included, [$NH_4^+$]:[$NO_2^-$] is about 1:10, and [$NO_2^-$] is overestimated. When yields and affinities are assumed identical for AOO and NOO, using AOO parameters for both (giving them identical $R^*$s), peak [$NH_4^+$] and [$NO_2^-$] are identical, and [$NO_2^-$] is underestimated. With only a difference in yield or the uptake affinity, [$NH_4^+$]:[$NO_2^-$] is about 1:3 or slightly lower, as quantified above with the point balance, and consistent with observed [$NO_2^-$] and observed ratios[10,12,13]. This suggests that either the yield or the affinity difference, or a smaller

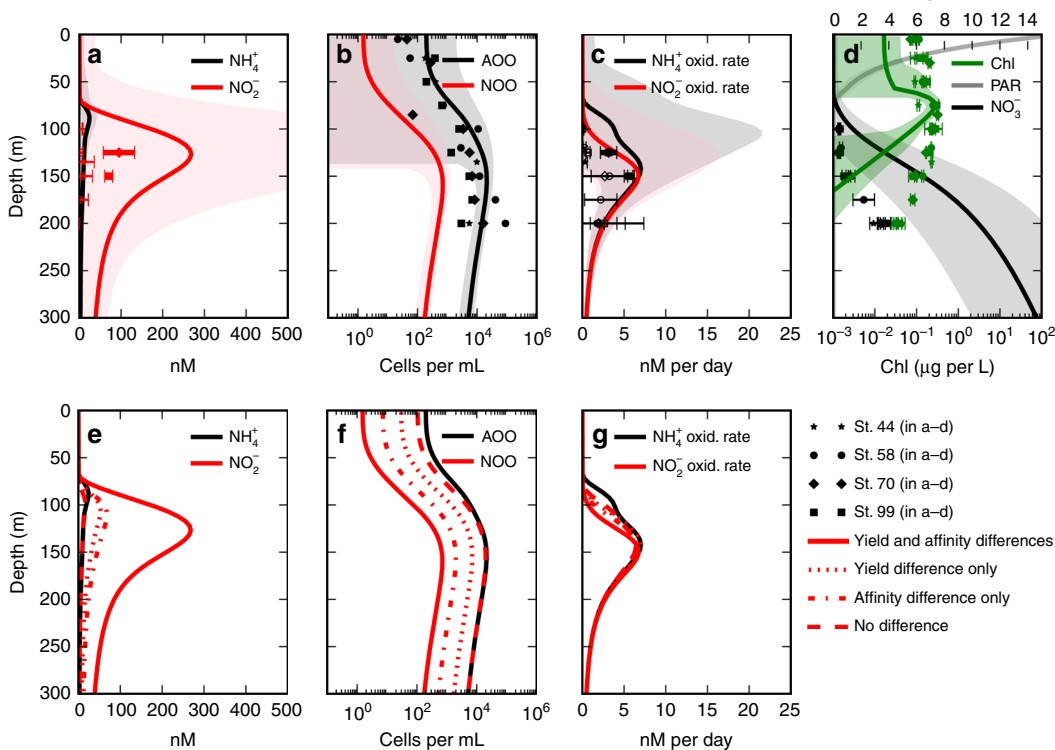

**Fig. 3** Stratified water column model solutions and observations from the subtropical North Pacific. Model solutions are indicated with lines, with shaded areas denoting 1 s.d. of the model ensemble. Observations from the four stations are indicated with marked points in **a**–**d**, with error bars denoting 1 s.d.: **a** [$NH_4^+$] and [$NO_2^-$] (the shaded region of [$NO_2^-$] reaches about 1000 nM), **b** ammonia-oxidizing and nitrite-oxidizing organism (AOO and NOO) abundances (observed *amoA* gene abundances), **c** nitrification rates, and **d** [Chl *a*], PAR (scaled to fit plot), and [$NO_3^-$]. Observations below the detection limit are indicated with open (vs. filled) markers. Also shown are solutions to additional model experiments (**e**–**g**), with only the difference in yield (reflecting the threefold energetic difference), with only the difference in affinity (here, as the 4.6-fold decrease related to a 10-fold larger NOO cell volume), and with no quantitative difference in the parameters describing AOO and NOO metabolisms

combination of both, may best represent natural assemblages of nitrifiers.

Even deeper in the water column, [$NO_2^-$] decreases, resembling the observed deep "tail" of the PNM[6]. Observed pelagic [$NO_2^-$] has been shown to be much lower (nanomolar)[13] than the deep concentration in the model here (tens of nanomolar). Assuming no other sinks for [$NO_2^-$], we hypothesize that deep [$NO_2^-$] concentrations still reflect the *R**s of the NOO, but of a diverse community. Affinities and efficiencies may vary with clades, such as those with maximum growth rates too low for survival at the PNM, or may be plastic in nature. A slower-growing, efficient, deep NOO clade could deplete [$NO_2^-$] to lower levels.

**Nitrifier abundances and nitrification rates.** Modeled nitrifier abundances also reflect the distinctions between the AOO and NOO (Fig. 3b, f). With both yield and affinity differences, NOO abundance is 30-fold lower than AOO. In the model experiment with only the yield difference, NOO abundance is threefold lower than AOO. This threefold difference in abundance is consistent with observations of AOO and NOO cell abundances: a fourfold difference in the abundances of $NH_4^+$-oxidizing marine group 1 (MG1) (*Thaumarchaea*) and $NO_2^-$-oxidizing *Nitrospina* in the California Current[19], a 1–5-fold lower abundance of *Nitrospina*-like bacteria compared to *amoA* gene and MG1 abundances in Monterey Bay[58], and a 1–4-fold difference at Station ALOHA[58]. Thus, observed abundances are quantitatively consistent with observed [$NH_4^+$]:[$NO_2^-$] in oligotrophic environments, providing

compelling evidence for the differences between AOO and NOO metabolisms in natural assemblages.

Rates of $NH_4^+$ and $NO_2^-$ oxidation are identical below 150 m depth for all model experiments (Fig. 3c, g). This matches the lack of consistent differences in observed rates[2], though differences in coastal waters have been documented[41,42]. Thus, differences in the modeled AOO and NOO emerge in nutrient distributions and nitrifier abundances, but not in subsurface nitrification rates. This reinforces our understanding that exported organic matter determines the rate of all steps in the sequence of remineralizing metabolisms below the euphotic zone[2,12,13,22].

**Vertical mixing affects the magnitude of the PNM.** Modeled [$NO_2^-$] is higher than *R** from the peak of the PNM to about 175 m depth. This indicates that vertical mixing is non-negligible there. In other model simulations, particularly in less stratified water columns, this effect is stronger, and DIN accumulates to concentrations much higher than *R** (Supplementary Fig. 4). In Fig. 3, vertical mixing sweeps cells away from their location of favorable growth so that the NOO cannot sustain a population large enough to draw down [$NO_2^-$] to their *R**, resulting in [$NO_2^-$] accumulation to a higher concentration (see quantitative analysis in Supplementary Note 2).

This influence of mixing at the model PNM is consistent with residence time analyses that find that older [$NO_2^-$] exists at the peak PNM[10,21]. Biological turnover will be slower if nitrifiers are unable to metabolize all of the [$NO_2^-$] at a particular location. We

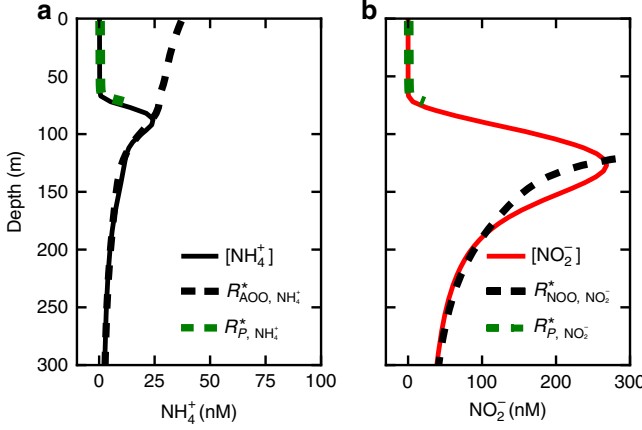

**Fig. 4** Water column model solutions and subsistence concentrations. **a**, **b** [$NH_4^+$] and [$NO_2^-$] with associated $R^*$s of ammonia-oxidizing and nitrite-oxidizing organisms (AOO and NOO) and picophytoplankton ($P$). The $R^*$s of $P$ here are calculated with the maximum light-limited growth rate (Eq. (28)), and thus become negative at depth once light renders photoautotrophy unsustainable (see Supplementary Fig. 3 for further details)

may thus consider two regimes within the PNM: a transport-influenced upper regime, and a nitrifier-controlled lower regime. This provides an interpretation of the double-peaked character of the PNM in some locations, as an alternative to the standing hypothesis that the upper PNM forms because of $NO_3^-$ reduction by photoautotrophs[9,18].

**Predicting global distributions of nitrite and nitrification**. Why does $NO_2^-$ accumulate at the subsurface in the subtropics, but also throughout the surface in subpolar regions in Fig. 1? To understand this distribution, we resolve the population dynamics of nitrifiers in a three-dimensional ocean circulation and biogeochemistry model. Building upon previous work[54,59], the model couples the cycles of nitrogen, carbon, phosphorus, iron, silica, and oxygen, includes both particulate and dissolved organic pools, and resolves multiple populations of phytoplankton and zooplankton functional types. We introduce the microbial nitrifying and heterotrophic types as described above. In the illustrated global simulation, the affinity of the NOO was increased (though still lower than AOO affinity; see Supplementary Table 1) to match the distribution of the maximum water column [$NO_2^-$] in the GLODAPv2 compilation (Fig. 5c).

The global model simulates the distribution of $NO_2^-$ in the transects from the GLODAPv2 database (Fig. 1)[60,61], predicting a PNM throughout the subtropics and the accumulation throughout the upper part of the water column poleward of about ±45°. What gives rise to this latter feature? At high latitudes, deep mixing transports fixed nitrogen (including $NO_2^-$) to the surface. If photoautotrophs are limited by light or iron, they do not deplete this DIN. Since the $R^*$ of the NOO is higher than that of the phytoplankton, surface $NO_2^-$ reflects this higher concentration.

In this case, surface $NH_4^+$ and $NO_2^-$ may be accessible to the $NH_4^+$- and $NO_2^-$-limited chemoautotrophs, and they can coexist with light- or iron-limited phytoplankton. The global simulation predicts this coexisting nitrification in the near-surface at high latitudes (Fig. 6b). In the model, this is mostly due to vertical mixing, as inferred from observations in the Southern Ocean[62,63]: DIN and nitrifier cells are swept up to the euphotic zone because

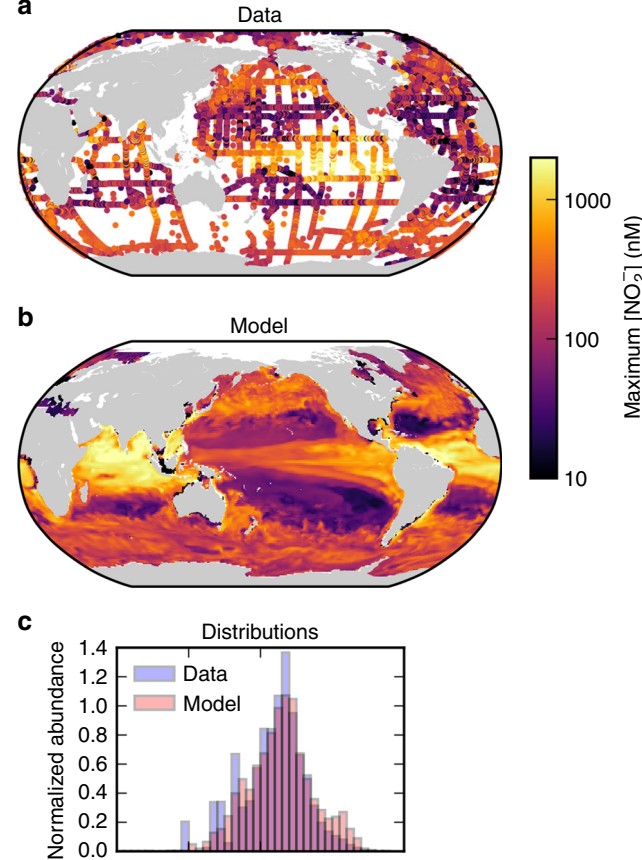

**Fig. 5** Observed and simulated maximum nitrite concentration in the oxygenated water column. **a** From the GLODAPv2 database[60, 61], **b** from the global model, and **c** the distributions of both. Only locations that co-occur with $O_2$ concentrations greater than 10 μM are plotted. (Note: a threshold of 50 μM $O_2$ results in visually indistinguishable versions of **a** and **b**.) Map generated with Python version 3.5.1, Matplotlib version 1.5.1, and Basemap version 1.0.7[86]

of deep mixed layers, and the nitrifiers coexist transiently with the phytoplankton. We diagnose that the nitrifiers can also coexist stably (locally) with the phytoplankton at some locations (Supplementary Fig. 5). Seasonal diagnostics of the global model reveal enhanced surface nitrification rates in the winter in both hemispheres as mixed layers deepen and phytoplankton growth becomes more light-limited (Supplementary Fig. 6). This is consistent with observations of increased *amoA* gene abundances and potential nitrification rates in the winter in surface waters in the coastal Arctic Ocean[36].

The simulated maximum water column nitrification rates correlate broadly with primary production (Fig. 6, Supplementary Fig. 7), since subsurface nitrification depends on surface production for substrate[2,10,11,13,22]. The range of maximum rates is consistent with the 10–100 nmol N L$^{-1}$ d$^{-1}$ range of compiled marine nitrification rate measurements[2]. Globally integrated $NH_4^+$ and $NO_2^-$ oxidation in the model are 3190 and 2460 TgN per year, respectively, and integrated primary production is 7150 TgN per year (40.4 PgC per year). As a global average, 17% of $NO_2^-$ oxidation is above 100 m depth in the model, providing a shallow source of $NO_3^-$ for phytoplankton. This is consistent with the understanding that nitrification

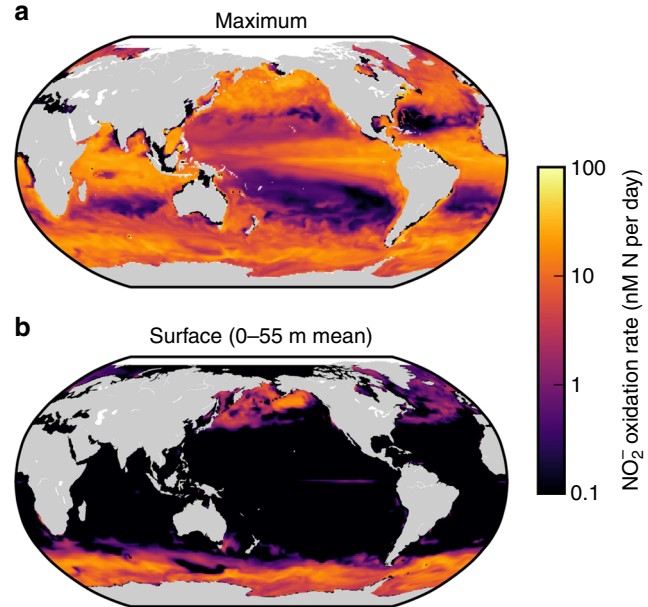

**Fig. 6** Simulated nitrite oxidation rate. **a** Water column maximum rate and **b** surface rate (0–55 m mean). Map generated with Python version 3.5.1, Matplotlib version 1.5.1, and Basemap version 1.0.7[86]

within the euphotic zone fuels a non-negligible fraction (here, 17%) of primary production[64].

In Fig. 5, we illustrate the maximum $[NO_2^-]$ in the water column where oxygen concentration is high (i.e., where $O_2 > 10\ \mu M$, which excludes the domain of anaerobic activity and the secondary $NO_2^-$ maximum). The model captures the lowest values in the subtropical gyres, and the higher values in high latitude and equatorial upwelling regions. The model does overestimate the values in the North Atlantic and Indian Equatorial regions. However, this overestimate disappears when nitrifiers are assumed to have the same temperature sensitivity as the other microbes (Supplementary Note 1). The model also underestimates the highest values in the Pacific Equatorial region. The coarse resolution of the physical model does not resolve the sharpness of equatorial circulation, although the climatology might also be biased by aggregating the effects of eddies, showing high $[O_2]$ and $[NO_2^-]$ at the same location that were not measured at the same time. Spatial patterns of mismatches between modeled and observed magnitudes could also indicate unaccounted-for diversity among the nitrifying community, which we discuss below.

## Discussion

We have developed a theoretical framework that predicts the locations of nitrification and $NO_2^-$ accumulation broadly. Competition with photoautotrophy explains why nitrification rates often peak at depth. When photoautotrophic phytoplankton and chemoautotrophic nitrifiers compete for $NH_4^+$ or $NO_2^-$, nitrifiers lose the competition because their metabolisms are much less efficient at using DIN for growth. This set of dynamics should characterize much of the N-limited surface ocean.

When phytoplankton growth is limited by something other than nitrogen (e.g. light or iron), nitrifiers may locally sustain growth if sufficient DIN is supplied. This characterizes the base of the euphotic zone, and subpolar surface environments during winter, where phytoplankton are light-limited. Surface nitrification rates may also be significant in high-nutrient, low-

chlorophyll regions where iron limits phytoplankton growth, if nitrifiers are less iron-limited. Observations suggest that this may be the case for AOO, which require copper rather than iron for redox machinery[3]. Additional model experiments that include an iron limitation to AOO and NOO growth show similar solutions (Supplementary Note 3), though a more thorough analysis of the iron requirements of nitrifying microorganisms is needed.

Future observations could test these hypotheses. While time-scales of marine nitrifier growth are too long for nitrification to become competitive at the surface at night (Supplementary Note 4, Supplementary Fig. 10), this could be tested in areas where phytoplankton have been light- or iron-limited for weeks or months at a time. Resource ratio theory[47] suggests that the degree of coexisting nitrification and primary production in the euphotic zone reflects the degree of limitation of phytoplankton growth. Less iron or less light availability should allow for a proportionally higher rate of coexisting nitrification.

Additionally, continual supply of $NH_4^+$, $NO_2^-$, and nitrifying biomass prevents competitive exclusion by phytoplankton and allows nitrifiers to sustain a transiently coexisting population. This may characterize a steady state near the surface in areas with sufficient vertical transport, such as when mixed layers are deeper than the euphotic zone or when upwelling is significant. This also may characterize coastal areas with high-nutrient injection from river runoff.

Thus, we suggest that light is an indirect control on nitrification in the water column. This can be reconciled with laboratory data showing direct light inhibition[26–28,30,39,40], if long-term exclusion from the surface has reduced the value of photo-protection and facilitated the evolution of photoinhibition in many clades of nitrifiers. We find that the base of the euphotic zone is an optimal location for nitrification in stratified water columns: nitrifying chemoautotrophy is outcompeted by photo-autotrophy above and limited by the availability of reduced DIN, sourced from the remineralization of organic matter by hetero-trophs, below.

We present two reasons for a distinction between the two steps of nitrification that allow for higher $[NO_2^-]$ than $[NH_4^+]$ in oligotrophic environments: (1) energetic constraints suggest that NOO require significantly more $NO_2^-$ to sustain their population relative to the amount of $NH_4^+$ required by AOO, and (2) allo-metry suggests that the specific nutrient affinity of NOO may be lower if they are on average larger than the AOO in a particular community, as recent work has suggested[50]. Both distinctions reduce the competitive strength of the modeled NOO, which cannot deplete $NO_2^-$ to as low of a concentration.

How well do these theoretical distinctions represent real nitrifying populations? The observed abundances of AOO:NOO and $[NH_4^+]:[NO_2^-]$ in oligotrophic environments are predicted by the distinction in yield alone. However, we know that organisms can develop enzymatic machinery to overcome physiological or energetic constraints. We expect a diversity of yields and affinities to characterize the marine nitrifying community. Here, growth efficiencies were estimated from marine batch cultures grown in initially nutrient-rich conditions, but efficiencies of cells may be significantly different across natural environments. Though the estimates included some mixotrophic cultures (Supplementary Note 5), other metabolic diversity may contribute to energy production[3,4], increasing yields. A contribution from "comam-mox" bacteria that completely oxidize $NH_4^+$ to $NO_3^-$ is now fathomable, given recent evidence that it may be oligotrophically adapted[65]: measureable $NO_2^-$-oxidation rates may be lower if comammox metabolizes a portion of the $NH_4^+$ pool. Even more speculatively, a yet-to-be-discovered small, high-affinity NOO population may also exist in the open ocean. Comparisons of marine AOO and NOO affinities and efficiencies could test these

hypotheses, and further connect the dots between the proposed mechanisms and natural assemblages.

Though the model here resolves only one bulk type of AOO and NOO each with fixed parameters, the framework developed may be useful for linking patterns of diversity among multiple types to biogeochemical patterns. Our model shows that DIN concentrations are sensitive to the assumptions of the hypothesized yield and affinity differences (Fig. 3e–g), and this sensitivity may be exploited to gain insight into the controls on nitrifier diversity. For example, observations show a shift in dominance from ammonia-oxidizing bacteria (AOB) to ammonia-oxidizing archaea (AOA), as well as to different clades of AOA, from coastal to more oligotrophic environments and with an increase in depth[10,13,19]. Since larger AOB have lower affinity for $NH_4^+$ than AOA[37], AOB should have a larger $R^\star$, which may contribute to an explanation for the observed decrease in $[NH_4^+]:[NO_2^-]$ across the productivity gradient of the California Current[10]. We note, however, that time-varying fluxes of $NH_4^+$ from faster growing organisms closer to the dynamic mixed layer make it less likely that the steady-state approximation should hold for $NH_4^+$, and so $[NH_4^+]$ may be less predictable than $[NO_2^-]$. Eddy circulation, not resolved in this global model, will also contribute spatial and temporal heterogeneity to the basin-scale patterns simulated here[66].

Even at steady state, factors other than the nitrifier metabolic parameters cause variation in the magnitudes of $[NH_4^+]$ and $[NO_2^-]$. In Eq. (3), the subsistence resource concentration depends also on the loss rate, which varies in space and time according to the population density of predators (or viral lysis). This top–down control, one of the most uncertain parameterizations in ecosystem models, could cause additional differences between subsistence concentrations. Different grazing parameterizations do not qualitatively affect model solutions, though they change the specific depth at which nitrification becomes sustainable. In addition, we found that the modeled PNM was higher than predicted because of vertical mixing. High rates of mixing can drive the actual resource concentration away from the subsistence resource concentration predicted by Eq. (3)[67].

Thus, AOO and NOO that have a potential to draw down DIN to very low concentrations (e.g., in batch cultures) can be associated with much higher DIN concentrations where losses and mixing are significant. Our models suggest that this is often the case. This highlights the utility of a dynamic ecosystem model in synthesizing complex interactions. Resolving the fluxes of all of the components results, at times, in unintuitive relationships between the standing stocks of nutrients and biomass.

In the current model, nitrification is sufficient to simulate global distributions of $NO_2^-$ without phytoplankton excretion of $NO_2^-$. Previous modeling suggests that excretion by phytoplankton at depth can also form the PNM[14]. Laboratory observations[23–25] and observed associations of *Prochlorococcus* with the PNM[68] support this mechanism. Such excessive $NO_3^-$ reduction by phytoplankton may also reflect the energetic constraints examined here, since the reverse of the sequence of nitrification redox reactions happens within phytoplankton cells. Here we do not present an argument against phytoplankton excretion also contributing to $NO_2^-$ accumulation except to point out that isotopic analysis suggests otherwise in some environments[10,21]. Rather, we provide explanations for $NO_2^-$ accumulation where nitrification contributes significantly to the $NO_2^-$ pool. Observations of nitrifier abundances and ambient nitrification rates are consistent with our hypotheses.

To conclude, we have hypothesized the main controls on the distributions of $NO_2^-$ in oxygenated waters and on the biogeography of nitrifying microorganisms. Additionally, we have

presented a dynamic parameterization of chemoautotrophic metabolisms suitable for global ocean biogeochemical models. The model articulates the rapid nitrogen cycling by microbial respiration at the base of the euphotic zone, which should aid in sharpening descriptions of export production. The model can also be extended to examine the interactions of anaerobic as well as aerobic nitrogen-cycling metabolic functional types within an aerobic–anaerobic microbial ecosystem, which may provide insight into the fate of fixed nitrogen in anoxic zones.

## Methods

**Theoretical nitrifier functional type development.** Following established methodology[48], three half-reactions combine to form the catabolic and anabolic full reactions for each nitrifier metabolism: (1) the oxidation of $NH_4^+$ or $NO_2^-$, (2) the reduction of oxygen, and (3) biomass synthesis. The parameter $f$ represents the fraction of electrons from the electron donor that are channeled into biomass synthesis vs. respiration. Combined, these reactions inform a "whole organism" stoichiometry. The resulting correlation of yield $y$ and growth rate $yV$ (Eq. (2)) thus characterizes the overall efficiency of particular metabolism (though tradeoffs between $y$ and growth rate may be important in modifying further characteristics).

We assume a fixed C:N biomass of $5 \pm 1$, as observed for heterotrophic marine bacteria[69], and that both AOO and NOO use reduced nitrogen (i.e., the same oxidation state as $NH_4^+$ and organic nitrogen) as the source of the elemental nitrogen for synthesis. This latter assumption represents the NOO population most realistically[4] and increases our burden of proof to distinguish the two metabolisms energetically, since the electron donor reaction is the sole energetic difference between them. For simplification, and following previous methodology[48], we here neglect the formation of $N_2O$ as a byproduct of $NH_4^+$ oxidation, which should have a negligible impact on AOO stoichiometry, given $N_2O$ yields per mol N nitrified of $< 1\%$[70].

For the $NH_4^+$ oxidizer (here considering $NH_4^+$ and $NH_3$ interchangeably), the three half-reactions, for generic biomass $C_cH_hO_oN_n$, and their electron-partitioning coefficients, are:

$$(1) \left[ \tfrac{1}{6}NH_4^+ + \tfrac{1}{3}H_2O \rightarrow \tfrac{1}{6}NO_2^- + \tfrac{4}{3}H^+ + e^- \right]$$
$$(1-f) \left[ \tfrac{1}{4}O_2 + H^+ + e^- \rightarrow \tfrac{1}{2}H_2O \right]$$
$$(f) \left[ \tfrac{n}{d}NH_4^+ + \tfrac{c-n}{d}CO_2 + \tfrac{n}{d}HCO_3^- + H^+ + e^- \rightarrow \tfrac{1}{d}C_cH_hO_oN_n + \tfrac{2c-o+n}{d}H_2O \right],$$

where $d$ normalizes the biomass synthesis reaction to one electron (see definition below). The sum gives the full metabolism for $NH_4^+$-oxidizing biomass $B_{AOO}$ (ignoring water and lumping bicarbonate into the $CO_2$ pool for simplification), as a function of $f$:

$$\left( \tfrac{1}{6} + \tfrac{f}{d} \right) NH_4^+ + \tfrac{cf}{d}CO_2 + \tfrac{1-f}{4}O_2 \rightarrow \tfrac{f}{d}B_{AOO} + \tfrac{1}{6}NO_2^-$$

For the $NO_2^-$ oxidizer, the three half-reactions are:

$$(1) \left[ \tfrac{1}{2}NO_2^- + \tfrac{1}{2}H_2O \rightarrow \tfrac{1}{2}NO_3^- + H^+ + e^- \right]$$
$$(1-f) \left[ \tfrac{1}{4}O_2 + H^+ + e^- \rightarrow \tfrac{1}{2}H_2O \right]$$
$$(f) \left[ \tfrac{n}{d}NH_4^+ + \tfrac{c-n}{d}CO_2 + \tfrac{n}{d}HCO_3^- + H^+ + e^- \rightarrow \tfrac{1}{d}C_cH_hO_oN_n + \tfrac{2c-o+n}{d}H_2O \right]$$

which when summed gives the full metabolism $NO_2^-$-oxidizing biomass $B_{NOO}$ as:

$$\tfrac{1}{2}NO_2^- + \tfrac{f}{d}NH_4^+ + \tfrac{cf}{d}CO_2 + \tfrac{1-f}{4}O_2 \rightarrow \tfrac{f}{d}B_{NOO} + \tfrac{1}{2}NO_3^-,$$

where the requirement of one mole of $NH_4^+$ per mole NOO biomass is effectively negligible in all model simulations.

The resulting nitrifier yields $y$, defined as moles biomass N synthesized per mole DIN used, are:

$$y_{NH_4} = \frac{1}{1 + \frac{d}{6f}} \approx \frac{6f}{d} \, (\text{for small} f), \tag{6}$$

$$y_{NO_2} = \frac{2f}{d}. \tag{7}$$

Since we assume that the $NO_2^-$ oxidizer also uses reduced nitrogen, we use the same estimate for $d$ for both functional types. Following previous methodology, $d$ represents the number of electron equivalents that correspond to the oxidation states of the inorganic constituents of that synthesis[48]. Assuming generic microbial biomass composition of $C_5H_7O_2N$ and $d = 4(5) + 1(7) - 2(2) - 3(1)$ gives $d = 20$.

**Estimate of nitrifier efficiency from data**. The whole-organism stoichiometries require a constraint on the fraction of electrons donated to biomass synthesis vs. respiratory energy production ($f$). Smaller $f$ equates to a lower growth efficiency and a lower yield, $y$. While thermodynamics can provide some theoretical constraints on $f$, the efficiency of energy production in marine microbes is not well known. Thermodynamics and wastewater treatment studies suggest that $f$ is very similar for the two nitrifying types[48]—the difference in the free energy of the oxidation of $NH_4^+$ to $NO_2^-$ compared to $NO_2^-$ to $NO_3^-$ (+41.65 and +32.93 kJ e$^-$ eq$^{-1}$, respectively, at standard state with pH = 7.0[48]) does not significantly impact theoretical estimates—so we employ published laboratory observations of marine AOO and NOO growth to constrain a common $f$ for both metabolisms[37,49,71–73] (Supplementary Table 2).

Yields $y_{NH_4}$ and $y_{NO_2}$ were estimated from observations of cell growth on $NH_4^+$ or $NO_2^-$, and the value of $f$ was inferred for each using Eqs. (6) and (7) (Supplementary Table 2). Some of the observed growth was mixotrophic, which exhibited yields about 10–20% higher than obligate chemoautotrophic growth (Supplementary Note 5). When required, the yield calculations assumed nitrogen cell quotas of 0.12 and 1.2 fmol N per cell for the AOO and NOO groups, respectively, with a range of 0.07–0.16 and 0.7–1.6 fmol N per cell contributing to uncertainty in the yields. These quotas are computed from the 10.2 ± 1.1 fg protein per cell content of AOA, as measured[37], and an assumption of a 10-fold larger quota for the NOO, based on the measured minimum 10-fold difference in protein content between AOA and AOB[37] and an assumed similar protein content for AOB and NOB. The nitrogen content of protein was assumed to be 16% by weight, and additional uncertainty was incorporated by considering a range of 10–20% (Supplementary Table 2), giving the above nitrogen quotas. The NOO nitrogen quota can also be independently estimated from the sphereoidical volume of a marine strain of *Nitrospina* of size 0.3–0.4 μm × 1–3 μm[49]: converting from an average bacterial carbon quota[74] of 0.22 g C cm$^{-3}$ with a C:N of 5 gives a quota of order 1 fmol N per cell, consistent with our estimate.

Our analysis supports the assumption that $f$ is similar for marine NOO and AOO populations in aggregate, though it ranges widely for NOO, and results in an average value on the order of 0.03. For the AOO and the NOO groups, the average yields are $(112 ± 32)^{-1}$ mol biomass N per mol $NH_4^+$ oxidized, and $(310 ± 320)^{-1}$ mol biomass N synthesized per mol $NO_2^-$ oxidized, respectively. The average $f$ values corresponding to these yields are 0.030 and 0.032, respectively. A value of 0.03 is about one-fifth of the value of $f$ inferred for wastewater bioreactors[48], perhaps explained by the need for marine organisms in oligotrophic environments to have higher-affinity, energetically-expensive nutrient transport systems.

The stoichiometries presented assume $f = 0.03$, $d = 20 ± 4$, with uncertainty in $d$ from the 1 mol/mol s.d. of the C:N[69], where changes in H and O stoichiometry were neglected to give the largest impact on the resulting stoichiometric uncertainty. We examine the sensitivity of the yields to variation in $f$ and $d$ in Supplementary Note 6 (Supplementary Fig. 11).

**Nitrifier uptake kinetics**. Kinetics experiments with cultured $NH_4^+$-oxidizing archaea *Nitrosopumilis* provide values for the parameters for the uptake of $NH_4^+$ by AOO[37] (Supplementary Table 1), including the information needed to convert to maximum specific uptake rate $V_{max}$ (in mol $NH_4^+$ per mol biomass N per day). The model incorporates a conversion from the reported maximum[37] $NH_4^+$ uptake of 24.2 ± 2.23 μmol $NH_4^+$ per mg protein per hour at 30 °C (Supplementary Table 1) with an associated half-saturation concentration of 133 ± 38 nM $NH_4^+$. Although a maximum rate of 51.9 μmol $NH_4^+$ per mg protein per hour was reported in batch culture, the growth of the cells was impaired by agitation, and so we do not expect this maximum rate to represent the maximum rate of cells in the ocean. The s.d. of the maximum rate was estimated from the s.d. of the corresponding oxygen uptake (36.29 ± 3.35 μmol $O_2$ per mg protein per hour: (3.35 × 24.2)36.29$^{-1}$ = 2.23). Assuming a 16% N content of protein gives a specific maximum $NH_4^+$ uptake rate of 50.8 ± 4.68 mol $NH_4^+$ per mol biomass N per day at 30 °C, which is the value of $V_{max}$ we use in the model. (See Supplementary Note 1 for discussion of temperature sensitivity.)

This maximum uptake rate corresponds to a maximum per cell nitrification rate of 5.92 ± 0.84 fmol $NH_4^+$ per cell per day (at 30 °C), assuming the measured cell quota for AOA[37] of 10.2 ± 1.1 fg protein per cell per day. This maximum rate is consistent with the measured cellular nitrification rates of enriched cultures of AOA[72,75] of about 2 and 2–4 fmol N per cell per day.

Measurement of the kinetics of natural assemblies of marine $NH_4^+$ oxidizers[29] show a lower half-saturation constant than *Nitrosopumilis* (27.2 ± 4.4 vs. 133 ± 38 nM $NH_4^+$) with respect to a bulk $NH_4^+$ oxidation rate of 24.9 ± 1.3 nM N d$^{-1}$. We would not expect all measured half-saturation constants to be identical, because they vary with their associated maximum rate, which is why the specific affinity ($V_{max}K_N^{-1}$) is the relevant trait[43]. However, we can infer from this comparison that natural assemblages may have a lower maximum rate, and thus that the model here may overestimate nitrification rates in some locations.

**Allometric theory and affinity**. Known $NO_2^-$-oxidizing bacteria are significantly larger in size than marine $NH_4^+$-oxidizing archaea[37,49,50]. We assume the 10-fold difference in volume discussed above. (Note: Recently, observations have suggested almost a fourfold larger cell diameter between dominant AOO and NOO types, equating to a 50-fold larger cell volume[50]). We then use established empirical and

theoretical allometric relationships[51,52] to predict the kinetic parameters for the NOO relative to those of the AOO. Allometric theory predicts that though the cellular uptake rate should increase with cell size, the specific uptake rate should decrease due to a decrease in the surface to volume ratio: the cellular rate scales with surface area as cell radius $r^2$, volume increases as $r^3$, and so the specific rate scales as $r^{-1}$. Theory also predicts that the half-saturation concentration increases with cell size. The diffusion-limited cellular uptake rate (which explains the steep slope of the Michaelis–Menton form) increases as $r$. The cellular affinity equates to the quotient of the cellular uptake rate and $K_N$. This suggests that $K_N$ scales as $r^2r^{-1} = r$. Together, allometry thus suggests that the specific affinity decreases with cell size as $r^{-2}$, which is supported by more detailed analysis of nutrient uptake models[43].

The 10-fold larger volume equates to a cell radius of NOO larger than that of AOO by $10^{1/3} = 2.2$, and so for the NOO, we estimate a specific uptake rate of about half, and a $K_N$ of about double that of the AOO. This gives a 4.6-fold lower specific affinity ($V_{max}K_N^{-1}$) of the NOO relative to the AOO.

Empirical results linking cell size to affinity[52] show that the affinity of AOA for $NH_4^+$ is of the same order of magnitude for that of picophytoplankton such as *Prochlorococcus*. For the AOA, using the above specific uptake rate of 50.8 mol $NH_4^+$ per mol biomass N per d and the 133 nM half-saturation concentration, the specific affinity is 382 L per μmol biomass N per d. In comparison, the specific affinity for a *Prochlorococcus* for a cell of diameter 0.6 μm, the average diameter of the *Prochlorococcus*[55], is of order 100 (calculated from allometric relationships[52] as $V_{max}(Q_{min}K_N)^{-1} = 88.2$ L per μmol biomass N per d). This literature[52] also suggests that picophytoplankton affinity for $NH_4^+$ in particular may be up to an order of magnitude higher than this: of order 1000. In sum, the theoretical specific affinity for a *Prochlorococcus*-sized cell for inorganic nitrogen is in the range of 100–1000 L per μmol biomass N per d, bracketing that of the nitrifiers. Thus, couched only in terms of affinity for $NH_4^+$ with respect to uptake (not growth), *Prochlorococcus* and AOA may be close competitors.

**Measurements**. Measurements were made in the subtropical North Pacific on NEMO ("Nutrient Effects on Marine Microorganisms") Cruise NH1417 in August and September 2014. Using satellite data, stations thought to be at productive locations were chosen for nitrification rate measurements to increase chances of detectable concentrations and rates. Measurements were taken at stations 44, 58, 70, and 99 (Supplementary Fig. 1; Supplementary Fig. 8). $NO_2^-$ and $NO_3^-$ concentrations were measured using standard colorimetric techniques[76], using a Varian Cary 100 Bio ultraviolet-visible spectrophotometer. Ammonia oxidation was measured using $^{15}NH_4^+$ as a tracer. Triplicate samples were spiked with enriched $NH_4^+$ tracer ($^{15}NH_4Cl$, 99%, Cambridge Isotope, 100 nM) incubated for 24 h in 500 mL polycarbonate bottles in the dark and at close to in situ temperatures. A 30 mL sample at the beginning and at the end of the incubation (T0 and T24) was extracted from each bottle, filtered through 0.2 μm pore-sized nylon filters, and frozen. Thawed aliquots were treated with sodium azide to convert all of the $NO_2^-$ to $N_2O$ gas[77]. The $^{14}N:^{15}N$ ratio of the $N_2O$ gas was analyzed on an isotope ratio mass spectrometer by the University of California Davis Stable Isotope Facility. Unlabeled carrier $NO_2^-$ (to 1 μM) from laboratory stock solution was added to reach instrument detection limits. A second treatment using $^{15}NO_2^-$ to measure the rate of $NO_2^-$ oxidation was unsuccessful because incomplete reduction of the remaining in situ spiked $NO_2^-$ inhibited accurate measurement of the enriched $NO_3^-$ pool (which does not mean that $NO_2^-$ oxidation rates were zero). The isotopic composition of the original samples was computed from the measured isotope ratio using a mass balance with the known carrier $NO_2^-$ concentration and isotopic composition. Ammonia oxidation rate (AOR; nM d$^{-1}$) was calculated as a function of the atom % $^{15}N$ ($a$) of initial and final samples as:

$$\mathrm{AOR} = \frac{a_{NO_2^-f} - a_{NO_2^-i}}{a_{NH_4^+}}[NO_2^-]_f(V\Delta t)^{-1}, \qquad (8)$$

where $a_{NH_4^+}$ is the atom percent of the $NH_4^+$ pool after spiking ($a_{NH_4^+} = a_{NH_4^+ \, spiked} - a_{NH_4^+ i}$, where $a_{NH_4^+ i}$ is the measured average of background samples), $V$ is the sample volume (L), and $\Delta t$ is the incubation time (d).

For the Chlorophyll $a$ (Chl $a$) measurements, seawater was filtered onto 25 mm Whatman GF/F filters (0.7 μm nominal pore size). Filters were extracted in the dark in 5 mL of 90% acetone for 24 h at +3 °C prior to measurement[78] on a Turner Designs TD-700 fluorometer calibrated with pure Chl $a$ (Sigma-Aldrich). Abundances of thaumarchaeal ammonia monooxygenase aubunit A (*amoA*) genes were measured with quantitative PCR.

**Ecosystem model**. Nine state variables are resolved as concentrations of nitrogen: the biomass of five functional type populations (ammonia-oxidizing organisms $B_{AOO}$, nitrite-oxidizing organisms $B_{NOO}$, phytoplankton $P$, heterotrophic bacteria $B_{het}$, and microzooplankton grazer $Z$), three inorganic nutrients ($NH_4^+$, $NO_2^-$, and $NO_3^-$), and organic detritus $D$. Total nitrogen is conserved in sum over the domain. Supplementary Table 1 lists all parameters, their dimensions, and the default values used in the model. The equations for the nine state variables are as follows, with the substantial derivative notation D/D$t$ including the diffusive flux as function of the diffusive coefficient $K$ as $\nabla \cdot (\vec{K}\nabla C)$ for tracer $C$, and advective fluxes as functions

of velocity $\overline{u}$ as $\nabla \cdot (\vec{u}C)$:

$$\frac{DNH_4^+}{Dt} = -\frac{1}{y_{NH_4}}\mu_{AOO}B_{AOO} - \mu_{NOO}B_{NOO} - V_{NH_4}P + \left(\frac{1}{y_D}-1\right)\mu_{het}B_{het}, \qquad (9)$$
$$+ (1-\zeta)gZ(P + B_{het} + B_{AOO} + B_{NOO})$$

$$\frac{DNO_2^-}{Dt} = \left(\frac{1}{y_{NH_4}}-1\right)\mu_{AOO}B_{AOO} - \frac{1}{y_{NO_2}}\mu_{NOO}B_{NOO} - V_{NO_2}P, \qquad (10)$$

$$\frac{DNO_3^-}{Dt} = \frac{1}{y_{NO_2}}\mu_{NOO}B_{NOO} - V_{NO_3}P, \qquad (11)$$

$$\frac{DD}{Dt} = -\frac{1}{y_D}\mu_{het}B_{het} + m_B(P + B_{het} + B_{AOO} + B_{NOO}) + m_ZZ^2 - \frac{\partial(w_sD)}{\partial z}, \qquad (12)$$

$$\frac{DB_{het}}{Dt} = B_{het}(\mu_{het} - m_B - gZ), \qquad (13)$$

$$\frac{DB_{AOO}}{Dt} = B_{AOO}(\mu_{AOO} - m_B - gZ), \qquad (14)$$

$$\frac{DB_{NOO}}{Dt} = B_{NOO}(\mu_{NOO} - m_B - gZ), \qquad (15)$$

$$\frac{DP}{Dt} = P(\mu_P - m_B - gZ), \qquad (16)$$

$$\frac{DZ}{Dt} = \zeta gZ(P + B_{het} + B_{AOO} + B_{NOO}) - m_ZZ^2. \qquad (17)$$

*Phytoplankton growth.* Phytoplankton grow as a function of a maximum growth rate $\mu_{max}$ (d$^{-1}$), with limitation by nutrients ($\gamma_N$), modification by temperature ($\gamma_T$), and limitation by light. Light limitation was parameterized using an exponential form as a function of the instantaneous photosynthetic rate $\Gamma$ (d$^{-1}$) and the Chl $a$ to Carbon ratio $\theta$ (g/g)[53,79] as:

$$\mu_P = \mu_{max}\gamma_N\gamma_T\left(1 - \exp\left(\frac{-\theta}{\mu_{max}\gamma_N\gamma_T}\right)\right). \qquad (18)$$

Photosynthetic rate $\Gamma$ was computed as a function of photosynthetically active radiation $I$, the maximum quantum yield of carbon fixation $\phi$ (mol C mol$^{-1}$ photons), and the absorption of light by phytoplankton $a_{phy}^{chl}$ (m$^2$ (mgChl)$^{-1}$) representing a mean value over all wavelengths, as:

$$\Gamma = \phi a_{phy}^{chl}I. \qquad (19)$$

The Chl:C $\theta$ varies with photoacclimation, and is computed using a steady-state solution[79] as:

$$\theta = \frac{\theta_{max}}{1 + \frac{\Gamma\theta_{max}}{2(\mu_{max}\gamma_N\gamma_T)}}, \qquad (20)$$

where $\theta_{max}$ is a maximum ratio. In the water column model, $\theta$ is allowed to reach an arbitrary minimum value of $0.1\theta_{max}$, which does not affect solutions of any state variables, but does set the minimum modeled Chl $a$ concentration in Fig. 3.

Nutrient limitation is a function of the total concentration of all species of DIN:

$$\gamma_N = \frac{NH_4^+}{NH_4^+ + K_{NH_4P}} + \frac{NO_2^-}{NO_2^- + K_{NO_2P}} + \frac{NO_3^-}{NO_3^- + K_{NO_3P}}. \qquad (21)$$

The inhibition of $NO_2^-$ and $NO_3^-$ assimilation in the presence of $NH_4^+$ had a negligible effect in the water column model solutions, and so was not included (though it is included in the global model). The specific rates of uptake $V$ (d$^{-1}$) of each DIN species by the phytoplankton type are resolved as:

$$V_{NH_4} = \mu_P\left(\frac{\frac{NH_4^+}{NH_4^+ + K_{NH_4P}}}{\frac{NH_4^+}{NH_4^+ + K_{NH_4P}} + \frac{NO_2^-}{NO_2^- + K_{NO_2P}} + \frac{NO_3^-}{NO_3^- + K_{NO_3P}}}\right)$$

$$V_{NO_2} = \mu_P\left(\frac{\frac{NO_2^-}{NO_2^- + K_{NO_2P}}}{\frac{NH_4^+}{NH_4^+ + K_{NH_4P}} + \frac{NO_2^-}{NO_2^- + K_{NO_2P}} + \frac{NO_3^-}{NO_3^- + K_{NO_3P}}}\right).$$

$$V_{NO_3} = \mu_P\left(\frac{\frac{NO_3^-}{NO_3^- + K_{NO_3P}}}{\frac{NH_4^+}{NH_4^+ + K_{NH_4P}} + \frac{NO_2^-}{NO_2^- + K_{NO_2P}} + \frac{NO_3^-}{NO_3^- + K_{NO_3P}}}\right)$$

Values for the maximum growth rate and the half-saturation constants were computed as functions of cell size following data-based allometric relationships[52]. Cell volume $v$ was converted from diameter assuming a spherical cell, and traits calculated with generic form:

$$\mu_{max} = av^b. \qquad (22)$$

For the picophytoplankton functional type ($P$ in the water column model), $a$ and $b$ are chosen for small cells[80], with $v$ computed assuming the average *Prochlorococcus* cell diameter of 0.6 μm[55]. For the larger additional phytoplankton types in the global model, $a$ and $b$ are representative of larger phytoplankton cells using published values[52,54].

The effective half-saturation constants for DIN uptake with respect to $\mu_{max}$ were calculated by conversion from the allometric relationships[52] for half-saturation constants $K_N$ with respect to maximum uptake rate $V_{max}$ and minimum cell quota $Q_{min}$ as[57]:

$$K_{NO_xP} = K_N(v)\frac{\mu_{max}(v)Q_{min}(v)}{V_{max}(v)} \qquad (23)$$

with published values of $a$ and $b$ for uptake and $Q_{min}$[52]. The half-saturation constant for $NH_4^+$ was assumed to be half that of the more oxidized species: $K_{NH_4P} = 0.5K_{NO_xP}$. Final values for all model parameters are listed in Supplementary Table 1.

*Heterotrophic bacterial growth.* The bacterial heterotrophic functional type grows as a function of organic matter (detritus $D$) as:

$$\mu_{het} = y_DV_{maxD}\frac{D}{D + K_D}\gamma_T \qquad (24)$$

where $y_D$ partitions consumption of $D$ into biomass synthesis (as $y_D$) and its remineralization into $NH_4^+$ (as $y_D - 1$). The growth efficiency $y_D$ is assumed as the average bacterial growth efficiency of 0.14 (s.d. 0.14) (mol biomass synthesized mol $D^{-1}$) for the open ocean[81]. The maximum uptake rate $V_{maxD}$ and half-saturation constant $K_D$ are best estimates that constrain the model heterotrophic bacterial growth rate to about 0.1 d$^{-1}$, matching the average bulk bacterial growth rate[82].

*Grazing.* A single zooplankton grazer consumes picophytoplankton (solely the *Prochlorococcus*-like type in the global model), heterotrophic bacteria, and the chemoautotrophic nitrifiers. This parameterization was chosen based on the assumption that grazing preferences are predominantly governed by size[83], and that *Prochlorococcus* and the microbial functional types are all roughly the same size. (Thus, there is a possibility that differences in size between AOO and NOO may influence this top–down control.) The total amount of grazing is calculated as a saturating function of total prey biomass, giving the rate of grazing $g$ (L mol$^{-1}$ d$^{-1}$), which is then multiplied by the biomass of each prey in Eqs. (13–16) and summed for total consumption by $Z$ in Eq. (17), as:

$$g = g_{max}\frac{1}{P + B_{het} + B_{AOO} + B_{NOO} + K_g}\gamma_T \qquad (25)$$

with maximum grazing rate $g_{max}$ and half-saturation $K_g$. Though these values are uncertain, a recent compilation[83] suggests that the two values are of the same order of magnitude, constraining their influence on $g$. Here, we assume $g_{max} = K_g = 1$, which is of the order of magnitude for these values used in previous marine ecosystem models[54].

The zooplankton grazer also excretes $NH_4^+$ via respiration as governed by its growth efficiency $\zeta$. Studies of zooplankton nitrogen growth efficiency is variable, and that 0.5 is a reasonable mid-range value[83].

*Temperature.* Other than the nitrifier growth[56], all microbial growth, grazing, and mortality rates are represented as a function of temperature (non-dimensional $\gamma_T$)

using a formulation that follows the Arrhenius equation[54] as:

$$\gamma_T = \tau \exp\left(A_E\left(\frac{1}{T + 273.15} - \frac{1}{T_0}\right)\right), \quad (26)$$

where $T$ is the ambient temperature, $T_0$ is a reference temperature, $A_E$ regulates the temperature modification, and $\tau$ normalizes the maximum value. (See Supplementary Note 1 for discussion of temperature sensitivity).

**Calculation of $R$*s.** $R$*s are calculated with Eq. (3) for AOO and NOO, where $y_{ij}V_{\max_{ij}}$ is the maximum growth rate. For phytoplankton $R$*s, the maximum growth rate is considered to be its maximum light-limited growth rate $\mu_{\text{light}}$, from Eq. (18) without the limitation by nutrients, as:

$$\mu_{\text{light}} = \mu_{\max}\gamma_T\left(1 - \exp\left(\frac{-\theta}{\mu_{\max}\gamma_T}\right)\right), \quad (27)$$

which is acquired from the steady-state solutions. Thus, $R$* for $P$ is calculated as:

$$R_{ij}^* = \frac{K_{ij}L_i}{\mu_{\text{light}} - L_i}. \quad (28)$$

For both phytoplankton and nitrifiers, the loss rate $L$ is calculated as the sum of respiratory or maintenance losses $m_B$ (d$^{-1}$) and the resulting steady-state $gZ$ (d$^{-1}$), as

$$L_i = m_B + gZ, \quad (29)$$

where $g$ (L mol$^{-1}$ d$^{-1}$) is the above grazing rate (equation 25) and $Z$ (mol L$^{-1}$) is the population density of predators. In the model here, the loss rate $L_i$ is the same for all of the microbial prey populations, since $m_B$ is assumed constant and $g$ depends on the sum of all prey biomass. (This specific rate $L_i$ is multiplied by the biomass of each prey $B_i$ in Eqs. (13–16) to give biomass-dependent losses.)

**1D water column physical environment.** In the water column model, the mixed layer was imposed by varying the vertical diffusion coefficient $K_Z$ with depth, from a maximum $K_{\max}$ at the surface to a minimum $K_{\min}$ with a length scale of $z_{\text{mld}}$. The fixed (no flux) boundary conditions result in some accumulation of $D$ at the bottom of the 2000 m domain, conceptually representing a sediment layer. To smooth over numerical error, vertical mixing was allowed to increase there with a 100 m length scale, simulating a bottom boundary mixed layer. $K_Z$ (m$^2$ s$^{-1}$) is thus calculated at cell faces as:

$$K_Z = K_{\max}e^{-\frac{z}{z_{\text{mld}}}} + K_{\min} + K_{\max}e^{-\frac{z-H}{100}}, \quad (30)$$

where $H$ is the height of the domain (2000 m).

Light energy $I$ decreases with depth according to the attenuation coefficients for water $k_w$ and for chlorophyll $k_{\text{Chl}}$, following a previous approach[84] as:

$$I(z) = I_{\text{in}}e^{-\left(z\left(k_w + \sum_{n=1}^{z}(T_{\text{Chl}}(z)k_{\text{Chl}})\right)\right)} \quad (31)$$

where $T_{\text{Chl}}$ is the sum of the concentrations of chlorophyll, and $k_{\text{Chl}}$ is an upper estimate of the absorption by chlorophyll to account for additional absorption by colored dissolved organic matter[54]. $I_{\text{in}}$ is the incoming irradiance, which is $0.5I_{\max}$, or for resolution of the daily cycle, $I_{\text{in}}(t) = 0.5I_{\max}(\cos(\omega t) + 1)$ where $\omega = 2\pi\, \text{d}^{-1}$.

A temperature curve was fit to the mean observations from the four stations sampled on Cruise NH1417 (Supplementary Fig. 9) with an exponential form:

$$T(z) = 12e^{z/150} + 12e^{z/500} + 2 \quad (32)$$

with temperature $T$ in °C.

The illustrated domain was 2000 m in height, with 5 m vertical resolution. Equations were integrated forward in time using the 4th order Runge-Kutta method until equilibrium (i.e., solutions independent of initial conditions). Advection for the sinking organic matter pool was carried out using the QUICK advection scheme, consisting of a linear interpolation between points weighted by an upstream second order curvature, resulting in third order accuracy. Fluxes were calculated at the faces of each grid cell, and concentrations at the centers.

**1D model ensemble with parameter uncertainty.** For the ~1000 model solutions in Fig. 3, we vary the values of the biological parameters in the illustrated model to communicate their uncertainty (see Supplementary Table 1 for the ranges of values). Values for the kinetic parameters of the nitrifiers were assigned randomly from the normal distributions from results of the oxygen kinetics experiments[37]. The relative larger size of the NOO as compared to AOO was varied from a factor of 1 to a factor of 20, thus, impacting the relative values of $V_{\max}$ and $K_N$ by a factor of $1^{1/3}$ to $20^{1/3}$. Because of their large uncertainty, the following parameters were sampled randomly over the linear range of ±50% of the default value: the maximum growth rate $\mu_{\max}$ and nutrient half-saturation constants $K_{NO_xP}$ for

phytoplankton (with $K_{NH_4P}$ computed accordingly); the parameters governing the consumption and respiration organic matter by heterotrophic bacteria: $V_{\max D}$, $K_D$, and $y_D$; the grazing parameters $g_{\max}$ and $K_g$, mortality rates $m_B$ and $m_Z$, and zooplankton efficiency $\zeta$.

**Global model.** The present configuration of the 3D MITgcm biogeochemical model resolves a total of six phytoplankton populations with parameters that represent the traits of the following six functional types: diatoms, picoplankton, diazotrophs, coccolithophores, and other large and other small phytoplankton. Four zooplankton types graze on the phytoplankton: one each specifically on the picoplankton (which also consumes the six introduced microbial types below), the other small phytoplankton, and coccolithophore types, and the fourth grazes on the diatom, the other large phytoplankton, and diazotroph type. The three-dimensional ocean circulation state estimate (the ECCO-GODAE state estimate) is from the configuration of the MITgcm as constrained by observations[85], and has a horizontal resolution of 1° × 1° and 23 levels of vertical resolution, from 10 m at the surface to 500 m at depth. The model was numerically integrated until rates of microbial activity equilibrated throughout the thermocline (200 years in the illustrated model).

In addition to the six phytoplankton types, six microbial metabolic functional types are included in the global ecosystem model, and are responsible for all organic matter remineralization, nitrification, and denitrification. Particulate and dissolved organic matter (POM and DOM) are consumed and subsequently remineralized by the aerobic heterotrophic bacterial functional type (as in the water column model), an anaerobic nitrate-reducing ($NO_3^- \rightarrow NO_2^-$) heterotrophic type, and an anaerobic denitrifying ($NO_2^- \rightarrow N_2$) heterotrophic type. The growth of each heterotrophic type is limited by the sum of POM and DOM, and POM and DOM are both taken up, weighted by the fraction of the limitation imposed by each as a function of the local concentration (analogous to the uptake of the three species of DIN by phytoplankton). Redfieldian C:N:P:Fe stoichiometries of bacterial types and demands are constant. The two aerobic nitrifier types (AOO and NOO) are included as in the water column model. A chemoautotrophic anammox functional type ($NH_4^+ NO_2^- \rightarrow N_2$) is also included. The stoichiometries for the three anaerobic metabolic functional types are described in Supplementary Note 7.

The depletion of oxygen and switch from aerobic to anaerobic respiration occurs dynamically in the model, following previous parameterization[45]. The lower organic matter yield for the anaerobic types results in the competitive exclusion of the anaerobic heterotrophs in oxygenated environments. The anammox type is likewise excluded from oxygenated environments by the aerobic nitrifiers. The assumed stoichiometries of anaerobic metabolisms results in [$NO_2^-$] greater than 1 μM, simulating a secondary $NO_2^-$ maximum, for which analysis is beyond the scope of the investigation here.

**Code availability.** Fortran code for the ecosystem model and the one-dimensional water column configuration is available at https://github.com/emilyzakem/eco-nitrify. The global model (the MITgcm) is available at http://mitgcm.org and the ecosystem component including the bacteria is available from git://gud.mit.edu/gud1.

**Data availability.** Supplementary Data 1 contains all data presented from Cruise NH1417.

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

## Acknowledgements

E.J.Z. thanks Elise Heiss and Silvia Newell for guidance with measurement strategy, Oliver Jahn for computational support, Andrew Babbin and Paul Berube for discussions, and the scientists and crew of Cruise NH1417 on R/V New Horizon for onboard support with measurements. Daniel Whitt processed and plotted the MODIS L2, AVISO, and ADCP data in Supplementary Fig. 1. We thank Brenner Wai and Eint Kyi for their help in sampling and analyses of *amoA* gene abundances, respectively. M.J.F. and S.D. are grateful for support from the Simons Collaboration on Ocean Processes and Ecology (SCOPE award #329108), the Gordon and Betty Moore Foundation (grant #3778) and NSF (grants OCE-1315201, OCE-1558702, and 1434007). Funding for contribution by M.J.C. came from SCOPE (award #329108) and NSF (grant OCE-1241263). M.M.M. was funded through NSF (grant OCE-1241093).

## Author contributions

E.J.Z., M.J.F. and S.D. wrote the paper. E.J.Z., M.J.F. and S.D. designed modeling research. E.J.Z., A.A.-H., S.Q.F. and R.W.F. developed nitrification measurement strategy. E.J.Z. executed modeling research and nitrification measurements. M.J.C. measured *amoA* abundances. M.M.M., G.L.D. and E.J.Z. measured Chl a concentrations, and M.M.M. and G.L.D. provided support with shipboard measurements.

## Additional information

**Competing interests:** The authors declare no competing interests.

