## [Peer Review File · Nature Communications]

Reviewers' comments:

Reviewer #1 (Remarks to the Author):

This is a nice study that uses theoretical ecology and physiological principles to explain the distribution of nitrite in the upper ocean. In general I like the approach and I think this will be a useful contribution to the field. I have one primary comment that I would like to see addressed before publication.

The authors have done interesting work to predict R^* of AOO and NOO based on redox chemistry and/or uptake kinetics. This all makes sense to me, but how this works in the full ecosystem model is less clear. In Fig. 4, R^* of AOO and NOO varies with depth, and in the text the authors say this is due to variation in grazing mortality with depth. This implies that there is a fixed grazing mortality rate at each depth, which can be used to calculate R^* . However, the ecosystem model does not have a fixed mortality rate, it has a dynamic grazer population. So, I would like to see more discussion of exactly what is being calculated here. Related - why does the autotroph R^* not vary with depth as well?

Also, in a simple consumer-resource model, the presence of the grazer would lead to top-down control of AOO/NOO biomass. Under these conditions, the DIN concentrations will still be affected by the traits of AOO/NOO, but also by the traits of the grazers, and the concentrations could be much different than what they would be in the absence of grazers. In the authors' ecosystem model there is a quadratic closure mortality term on the grazers, which leads to an intermediate situation where there is both bottom-up and top-down control on bacterial biomass. So, my question is whether the R^* values are close to what is predicted in the absence of grazing (i.e., top-down control is weak), or whether they are much higher, in which case the biology of the grazers becomes more relevant.

On the same topic, one thing that's confusing me is the peak in ammonium and nitrite in the 1D model. It makes sense that AOO and NOO are outcompeted by autotrophs in the N-limited photic zone. And it makes sense that biomass of AOO and NOO will tend to peak at the base of the photic zone, where they are not outcompeted and where reduced N has a high supply rate. And it makes sense that AOO and NOO have different R^* . But it's not clear why there should be a nitrite or ammonium peak near the base of the photic zone, based solely on R^* theory, because wherever AOO and NOO persist they will drive their respective resources down to the same R^* everywhere. So it seems grazers must be essential for allowing an increase in ammonium and nitrite concentrations. I think the manuscript and the ideas presented will be greatly strengthened by a discussion of these issues.

Reviewer #2 (Remarks to the Author):

General Comments:

The paper entitled, "Ecological control of nitrite in the upper ocean" is original, well written and influential piece of research that will move the field forward. The authors have constructed a biogeochemical model that can mimic the observations of nitrite and ammonium accumulation in the ocean based predominately on the physiological differences between ammonia oxidizing archaea, nitrite oxidizing bacteria and phytoplankton.

There are a couple of main issues I have in the paper. First, I am confused with the section on subsistence concentrations, R^* (starting at line 99). I am surprised that in the PNM the limiting resource is the nitrite or ammonium when this is the only in the area in the ocean that they

accumulate. I would think that the flux of ammonium and nitrite would be more important than the actual concentrations. Can you clarify how flux/standing stock affects the R^* ? Also, is the substrate concentration actually the limiting factor in these zones? I am not convinced that the population couldn't take nitrite to zero if it was limiting.

Also, the authors recognize there was a good amount of uncertainty in the model. This is clearly demonstrated in the large shaded regions of Figure 3. While I recognize all the reasons these uncertainties exist, I would like to see a description of how to move forward and what would help eliminate some of the uncertainty. For example are there certain areas of the ocean that would be more fitting for using this model than other parts?

Lastly, the authors end by stating that this model may provide insight into the fate of fixed nitrogen in anoxic regions. I think either this needs more explanation on how the model could do that or should be removed because I think it opens up an entirely separate complex discussion.

I think this paper deserves publication, and I applaud the authors for tackling a complex system and constructing a very sophisticated model that will be useful for numerous future studies.

Specific Comments:

Line 34: What does "each" refer to? Be more specific.

Line 57: Reference?

Line 62: "rooted in the metabolisms of the two distinct clades nitrifying microorganisms", what do you mean by this statement? Can you expand on this?

Line 64: Here you mention subpolar regions as areas where $[\text{NO}_2^-]$ is elevated in the surface, but it also accumulates at HNLC's (mentioned later), should you put both areas in this heading?

Lines 118-119: Confusing sentence, please expand.

Line 121: Here is where you introduce uptake affinity, maybe this should be presented in the earlier section, "Microbial population dynamics." In that section, you could spend more time explaining yield and affinity, since they come up many times in the discussion.

Line 123: What do you mean by stoichiometric and kinetic factors?

Line 126: How well do we know that NOO are larger than AOO? How large do these differences need to be to make an affect on the affinity?

Line 141: Could there be other limitations?

Line 149: I'm surprised that temperature does not affect growth rate of nitrifiers. Do you think this reference is applicable to all nitrifiers?

Figure 3a: Why does the model predict much higher $[\text{NO}_2^-]$ than the samples collected from the N. Pacific?

Line 194-195: Another place that I noted needing more explanation of yield vs. affinity.

Lines 222-227: I found this paragraph confusing. I would consider rewriting it and highlighting the main point.

Lines 238-240: I really like the idea of two PNM regimes.

Line 251: I don't think you've referred to Figure 5 before you mention Figure 6c. Those figures should maybe be reordered.

Line 261: What do you mean by a higher R^* organism? Is there a better descriptor you could use there?

Line 290: How did you choose 10uM? Ammonium and nitrite oxidation can occur at much lower oxygen concentrations (Bristow et al, 2016).

Line 300: Could nitrite be transported out of low oxygen regions and sediments into the oxygenated waters, as another mechanism to reduction on particles? Also, is there a need to address nitrite production on particles throughout the article and how it affects the model results?

Lines 320-324: Can you add excretion of nitrite by phytoplankton into the model? Would it change the results?

Line 336: Can you add more on how 'comammox' would fit into the observations/model?

Lines 374-376: How much data are there backing the size differences between nitrite oxidizing bacteria and archaea? Are their ranges in sizes in organisms depending on location? What locations may this affinity difference not exist?

Line 415: What is the result of neglecting nitrous oxide formation?

Line 437: Is this then assuming AOB and NOB have similar protein content? What if they are different?

Line 454: How representative is Nitrosopumilis from ref 41? Do you expect there to be major differences in the oceanic AOO.

Line 509: Why were the nitrite oxidation rate incubations unsuccessful? Does that mean there were zero rates?

Reviewer #3 (Remarks to the Author):

Nitrate (NO_3) is the prevailing form of fixed inorganic nitrogen in the (interior of the) ocean. Nitrification is the microbiological process converting ammonium (NH_4), which is the inorganic nitrogen species released during organic matter degradation, to NO_3 . The intermediate product of this reaction, nitrite (NO_2), is rarely observed in the ocean, except for a thin layer at the bottom of the euphotic zone, the primary nitrite maximum (PNM) and for the mixed layer of (sub)polar waters. Generations of marine biologists and marine chemists had heard about the PNM at University, and certainly many have discussed the various theories to explain the PNM that have been put forward.

Zakem et al. use ocean models which apply rules of resource competition theory (for the first time) to tackle this problem. To start with, the authors provide a careful analysis of the stoichiometry and energetics of the two major agents of nitrification, ammonium oxidising organisms (AOO) and nitrite oxidising organisms (NOO). This careful analysis forms the basis of the developed models. Increasing the complexity from a 0D case (their point balance solution), through a vertical 1D model to a global (3D) ocean model they carefully introduce the elements of resource competition theory needed for their case and the interactions between biological processes and physical ones (mixing).

The major findings are that the surface ocean's lack of nitrite in oligotrophic euphotic zones is a consequence of competitive exclusion of nitrifying organisms by phytoplankton (which can make use of all forms of inorganic nitrogen) and that, in turn, limitation of phytoplankton by either light or iron, e.g. at the base of the euphotic zone and in high latitudes, allows for co-existence of slow growing chemoautotrophic nitrifiers and phytoplankton. Energetics and stoichiometry of the two steps of nitrification explain details of the vertical zonation (PNM, NH_4 -maximum).

The paper is well structured and well written. I also liked the details provided in the methods and supplementary information, putting the reader into the position to judge the uncertainties of parameters and hence the model results. The authors are also careful to stress limitations of their modelling approach, i.e. that it can't rule out by itself processes which have not been modelled. The authors follow a state-of-the-art open access policy and provide the 1D model to reviewers and readers. This contributes not only to reproducibility of science but may also, after publication, stimulate readers to adopt elements of the approach of Zakem et al.

The application of resource competition theory to the problem of global nitrite distribution is novel, in particular in this explicit modelling strategy. The argumentation is clear and benefits from the strategy of increasing the complexity, see above. Further the work, in particular its theoretical approach, the

methodological developments and the availability of the code provide for a large likelihood that this publication will see follow up work, by the authors and others. One example is already obvious from the methods section, where the application of a variant of this model to questions of nitrogen loss in oxygen minimum zone (OMZ) somehow pops up. I see a very good potential that the application of resource competition theory, similar to what is demonstrated in this manuscript for nitrification, can help to disentangle the issue of anammox vs. heterotrophic denitrification (and all the other nitrogen conversions) in such waters.

After reading the manuscript for the first time I was unsure whether this work is topical enough, or interesting enough for researchers in related disciplines. Something necessary for a Nature journal, I guess. However, thinking about the potential of the approach described in this manuscript to improve also our understanding of OMZ nitrogen conversions, I am convinced the paper is interesting to many readers of Nature journals. And finally, this study will find many readers, i.e. those that thought about the hows and whys of the PNM many year ago, when they entered the field of ocean research.

I suggest acceptance of the ms by Nature Communications with minor/technical revisions (s. below).

A have a very few minor remarks:

Caption Fig. 3, panel c. 'Observations of [NO₂-] and NH₄⁺ oxidations rates ...', I guess it shall be '..of NO₂- and NH₄⁺ oxidations rates ...', right?

Lines 242-244: The initial sentences introducing the global model were a little irritating. I first wondered why you now move from 1D to 3D, why don't you simulate subpolar system with a 1D model here. I am not saying you should, but perhaps you could introduce the need for the 3D model not just with the subpolar problem.

Lines 289-302. This is the only paragraph which I found somewhat difficult to read, since the argumentation is a little winding. I suggest to leave out the 2nd sentence (290-293), then write: 'Both models(Fig. 6). However, this ...'. Finally, I was irritated by 'The discrepancy could ... (301-302), which discrepancy? At the beginning of the paragraph you pointed out that observations and model agree.

Fig. 6 caption: I suggest to mention also in the caption (like in the text) that is is the max of NO₂ for O₂>10.

Reviewer #3 Additional Comments:

Line 422: Could you provide a little more detail how you arrived at the equation $d=4c+h-2o-3n$

Line 468: References 74 and 75 after d^{-1} has to be given in different format.

Line 491-492: Closing bracket of '(calculated ...' is missing, I think.

Line 585: Advection for 1D model is not clear, in particularly since in the code it states 'No advection'

Supporting Material.

Section S1.2 Several times you to some text 'above' (line 23, 28, 29). It seems that this suppl text has been extracted from a differently structured document. Please adjust such that the context is better linked to, e.g. by pointing to the main text instead of 'above'.

Tab. S2 caption: Perhaps point out 'in the 1D-water column model'

Reviewers' comments:

We would like to thank all three of the reviewers for their very thoughtful and constructive comments. Based on these, the revised manuscript is much improved.

Reviewer #1 (Remarks to the Author):

This is a nice study that uses theoretical ecology and physiological principles to explain the distribution of nitrite in the upper ocean. In general I like the approach and I think this will be a useful contribution to the field. I have one primary comment that I would like to see addressed before publication.

The authors have done interesting work to predict R^* of AOO and NOO based on redox chemistry and/or uptake kinetics. This all makes sense to me, but how this works in the full ecosystem model is less clear. In Fig. 4, R^* of AOO and NOO varies with depth, and in the text the authors say this is due to variation in grazing mortality with depth. This implies that there is a fixed grazing mortality rate at each depth, which can be used to calculate R^* . However, the ecosystem model does not have a fixed mortality rate, it has a dynamic grazer population. So, I would like to see more discussion of exactly what is being calculated here. Related - why does the autotroph R^* not vary with depth as well?

We realized this wasn't clear and have provided a more complete discussion on the top down and bottom up controls of R^* . We have redone Figure 4 to be clearer, included a substantial better explanation, added explicit description of obtaining R^* in the methods, and an additional figure in the Supplemental. We explain and quote these changes below.

R^* is a diagnostic quantity as presented in Fig. 4: we solve the model numerically for steady state, and then use the resulting depth-varying loss rate (a function of the dynamic grazer population, see equations in methods) and depth dependent maximum growth for phytoplankton to provide the depth dependent R^* . In the one-dimensional results, the difference between exact solution from the R^* and the nutrient concentration is due to physical transport of the biomass (i.e., vertical mixing in the 1D model). The autotroph R^* varies with depth due to the grazing, but also to light. However, in the original Fig 4 the value was so low that the variance was not noticeable.

We now add a substantial longer explanation of how R^* s were calculated in the main text, and have included a section “**Calculation of R^* s**” in the Methods (**line 574**). For clarity, we have altered Fig. 4 to incorporate the depth-dependent maximum light-limited maximum growth rate for phytoplankton (which thus increases with depth, visually indicating the turnover from phytoplankton to nitrifier control). This light-limited maximum growth rate is defined in this new Methods section (equation 28).

We have also added Supplementary Figure 3, which (1) shows R^* s on a log scale, so that the depth-variance for all is perceivable, and (2) includes panel c that illustrates the depth-dependent L , Z , and the resulting growth rates for the P and the nitrifiers.

In main text (**line 158**): “**With the steady state solutions, we calculate the subsistence concentrations (R^* s) to diagnose ecological control (Fig. 4). We use equation (3) with y_{ij} V_{maxij} as the maximum growth rate for AOO and NOO, and equation (28) with the maximum light-limited growth rate for phytoplankton (see Methods). We then use the resulting steady state depth-varying L , a function of the dynamic grazer population Z**

(equation (29), Supplementary Fig. 3), to calculate the depth-dependent R^* s for NH_4 and NO_2 .”

Also, in a simple consumer-resource model, the presence of the grazer would lead to top-down control of AOO/NOO biomass. Under these conditions, the DIN concentrations will still be affected by the traits of AOO/NOO, but also by the traits of the grazers, and the concentrations could be much different than what they would be in the absence of grazers. In the authors' ecosystem model there is a quadratic closure mortality term on the grazers, which leads to an intermediate situation where there is both bottom-up and top-down control on bacterial biomass. So, my question is whether the R^* values are close to what is predicted in the absence of grazing (i.e., top-down control is weak), or whether they are much higher, in which case the biology of the grazers becomes more relevant.

Top-down control is an integral part of this ecosystem model, with loss rate L as directly proportional to the magnitude of R^* ; Thus we believe it is only relevant to discuss R^* in terms of including this loss rate. However, we believe that the new text (and additional methods and supplemental material) make this much clearer. This discussion also raises the issue of how sensitive the system is to the choice of grazing. Since parameterization of top-down control (i.e. grazing) is so uncertain, we were as “agnostic” as possible about it by parameterizing grazing pressure equally for all populations: one grazer consumes all populations with the same set of parameters. Thus, the loss rate is the same for AOO, NOO, and P (as now articulated in the Methods (“Calculation of R^* s”) and illustrated in panel c in Supplementary Fig. 3.) However, we also know that even this approach still involves a choice about how grazing affects community structure. We experimented with different grazing parameterizations (from quadratic mortality for the microbial populations and no explicit grazer Z , to multiple pools of specialist grazers). None of these parameterizations had an effect on the qualitative conclusions of this study – i.e. the PNM materializes as a switch from phytoplankton to nitrifier control of NO_2 , and NO_2 decreases below due to reduction in grazer control not matter what the assumptions of grazing were (though NO_2 concentration and depth of PNM could change depending on parameterization).

Because this is indeed an important point, we have added to the Discussion: **(line 343) “In equation (3), the subsistence resource concentration depends also on the loss rate, which varies in space and time according to the population density of predators (or viral lysis). This top-down control, one of the most uncertain parameterizations in ecosystem models, could also introduce other differences between the subsistence concentrations. Different grazing parameterizations do not affect qualitatively affect model solutions, though they change the specific depth at which nitrification becomes energetically favorable.”**

In the point balance section, we estimate the ratio of the R^* s, rather than their exact magnitudes. We have added a few words to this section to emphasize that we are calculating the ratio, and also now ask the question: **“What are the differences in R^* s..?” (line 101)** instead of “What are the values of R^* s..?” to clarify this.

On the same topic, one thing that's confusing me is the peak in ammonium and nitrite in the 1D model. It makes sense that AOO and NOO are outcompeted by autotrophs in the N-limited photic zone. And it makes sense that biomass of AOO and NOO will tend to peak at the base of the photic zone, where they are not outcompeted and where reduced N has a high supply rate. And it makes sense that AOO and NOO have different R^* . But it's not clear why there should be a nitrite or ammonium peak near the base of the photic zone, based solely on R^* theory, because wherever AOO and NOO persist they will drive their respective resources down to the same R^* everywhere. So it seems grazers must be essential for allowing an increase in ammonium and

nitrite concentrations. I think the manuscript and the ideas presented will be greatly strengthened by a discussion of these issues.

We believe that the new description of the R^* and the new version of Fig 4 make the ecological control on the peak clearer. At the base of the euphotic zone the NO_2 and NH_4 switch from being controlled by the phytoplankton (and its loss rates) to being controlled by NOO and AOO respectively. Since their R^* are so much larger (as is evident in Fig 4) than the phytoplankton R^* higher in the water column there is a surge in the concentration of NO_2 and NH_4 at this depth to match the new R^* control. However below this depth, reduced grazing leads to a decrease in R^* for the nitrifiers and hence lower concentrations. These two factors (switch from phytoplankton to nitrifier control and further reduction of grazing with depth) lead to a peak.

We point the reviewer to the new text (lines 163-185), Supplemental Fig. 3, and improved Fig. 4 to see that this is now better described. The cited lines in the text include the sentences: (line 180): **“The R^* s decrease with depth as grazing pressure becomes less (see blue dashed line in Supplementary Fig. 3c). Thus, after the increase, NH_4 and NO_2 then decline with depth, following the R^* predictions for the nitrifiers. Thus, subsurface maxima in both NO_2 (the PNM) and NH_4 emerge in the simulations. These maxima are controlled by both the top down and bottom up on the microbial community.”**

We have also added the following paragraph to the Discussion, which includes the above statement about top-down control (and also addresses comments from Reviewer #2):

(line 342) “Even at steady state, factors other than the nitrifier metabolic parameters cause variation in the magnitudes of NH_4 and NO_2 and specific locations of sustainable nitrification. In equation (3), the subsistence resource concentration depends also on the loss rate, which varies in space and time according to the population density of predators (or viral lysis). This top-down control, one of the most uncertain parameterizations in ecosystem models, could also introduce other differences between the subsistence concentrations. Different grazing parameterizations do not affect qualitatively affect model solutions, though they change the specific depth at which nitrification becomes energetically favorable. In addition, we found that $[\text{NO}_2]$ was higher than predicted because of vertical mixing. High rates of mixing can drive the actual resource concentration away from the subsistence resource concentration predicted by equation (3) (cite Levy). Thus, AOO and NOO that have a potential to draw down DIN to very low concentrations (e.g. in batch cultures) can be associated with much higher DIN concentrations where losses and mixing are significant. Our models suggest that this is often the case. This highlights the utility of a dynamic ecosystem model in synthesizing complex interactions. Resolving the fluxes of all of the components results, at times, in unintuitive relationships between the standing stocks of nutrients and biomass.”

Reviewer #2 (Remarks to the Author):

General Comments:

The paper entitled, “Ecological control of nitrite in the upper ocean” is original, well written and influential piece of research that will move the field forward. The authors have constructed a biogeochemical model that can mimic the observations of nitrite and ammonium accumulation in the ocean based predominately on the physiological differences between ammonia oxidizing archaea, nitrite oxidizing bacteria and phytoplankton.

There are a couple of main issues I have in the paper. First, I am confused with the section on subsistence concentrations, R^* (starting at line 99). I am surprised that in the PNM the limiting resource is the nitrite or ammonium when this is the only in the area in the ocean that they accumulate. I would think that the flux of ammonium and nitrite would be more important than the actual concentrations. Can you clarify how flux/standing stock affects the R^* ? Also, is the substrate concentration actually the limiting factor in these zones? I am not convinced that the population couldn't take nitrite to zero if it was limiting.

Ambient concentration of the limiting nutrients do indeed set growth rates of the microbes, but by consuming these nutrients the microbes (and their grazers) in fact also control the nutrient concentrations: they draw the nutrient down to the R^* (as given in Eqn. 3/Eqn 28). If nutrients were any lower, loss rates would be larger than growth and the microbes would not be able to exist. In a region of higher flux of nutrients the biomasses of the microbial populations and their grazers will be larger. As constructed (Eq 3), increased grazers will impact R^* and ambient nutrients will therefore be higher. It is this combination of top-down and bottom up controls on the R^* that sets the PNM, and we believe the new text highlights this much more clearly.

It is indeed potentially counter-intuitive that even in the PNM nitrifiers are NO_2 limited. But the theory (and model results) clearly show that this peak occurs when there is a shift of the control on NO_2 from phytoplankton to NOO – who have a significantly higher R^* for NO_2 (see updated Fig. 4). And less grazer control leads to lower R^* deeper in the water column. We believe that the revised text makes this argument significantly clearer – and acknowledges the importance of the top-down control. We also do further acknowledge the uncertainty in grazing parameters and parameterization.

The fuller description of the R^* control in the main text reads (lines 173-185):

Deeper in the water column, light becomes more limiting for phytoplankton, and the maximum light-limited growth rate decreases significantly, leading to the sharply increasing R^* s seen in Fig. 4. With still increasing depth, phytoplankton losses become larger than their growth rate (Supplementary Fig. 3c), and the phytoplankton can no longer survive (where the green dashed line ends in Fig. 4). At the point where phytoplankton are excluded, the nitrifiers become competitive for NH_4 and NO_2 , and the ambient nutrient concentrations are now set by their R^* s instead. This accounts for the large increase in $[\text{NH}_4]$ and $[\text{NO}_2]$ at about 100m (in this example).

The R^* s decrease with depth as grazing pressure becomes less (see blue dashed line in Supplementary Fig. 3c). Thus, after the increase, NH_4 and NO_2 then decline with depth, following the R^* predictions for the nitrifiers. Thus, subsurface maxima in both NO_2 (the PNM) and NH_4 emerge in the simulations. These maxima are controlled by both the top down and bottom up on the microbial community. As in the point balance, NO_2

accumulates to a higher maximum concentration than NH₄ because of the yield and affinity distinctions between AOO and NOO.”

Additionally, in the Discussion, we revise and add discussion of these dynamics:

(line 342) “Even at steady state, factors other than the nitrifier metabolic parameters cause variation in the magnitudes of NH₄ and NO₂ and specific locations of sustainable nitrification. In equation (3), the subsistence resource concentration depends also on the loss rate, which varies in space and time according to the population density of predators (or viral lysis). This top-down control, one of the most uncertain parameterizations in ecosystem models, could also introduce other differences between the subsistence concentrations. Different grazing parameterizations do not affect qualitatively affect model solutions, though they change the specific depth at which nitrification becomes energetically favorable. In addition, we found that [NO₂] was higher than predicted because of vertical mixing. High rates of mixing can drive the actual resource concentration away from the subsistence resource concentration predicted by equation (3) (cite Levy). Thus, AOO and NOO that have a potential to draw down DIN to very low concentrations (e.g. in batch cultures) can be associated with much higher DIN concentrations where losses and mixing are significant. Our models suggest that this is often the case. This highlights the utility of a dynamic ecosystem model in synthesizing complex interactions. Resolving the fluxes of all of the components results, at times, in unintuitive relationships between the standing stocks of nutrients and biomass.”

The addition of panel c to Supplementary Fig. 3 (explained above in response to Reviewer #1), emphasis how R* is calculated diagnostically as a function of loss rate (also cited above in response to Reviewer #1), helps in clarifying the nutrient control by the ecosystem.

Also, the authors recognize there was a good amount of uncertainty in the model. This is clearly demonstrated in the large shaded regions of Figure 3. While I recognize all the reasons these uncertainties exist, I would like to see a description of how to move forward and what would help eliminate some of the uncertainty. For example are there certain areas of the ocean that would be more fitting for using this model than other parts?

Thank you, yes, we agree more discussion is warranted. This is partially addressed in the new paragraph quoted above (line 342-365).

A major direction for moving forward would be in better understanding of the relative importance of affinity versus yield differences between AOO and NOO. We mention this now in several locations in the manuscript:

(lines 186-194): “We further demonstrate the control of NH₄:NO₂, and its sensitivity to the nitrifier parameters, with model experiments in which we isolate the differences in yields and affinities of the NOO and AOO (Fig. 3e--g). With both the yield and affinity differences between the nitrifiers included, NH₄:NO₂ is about 1:10, and NO₂ is overestimated. ... This suggests that either the yield or the affinity difference, or a smaller combination of both, may best represent natural assemblages of nitrifiers.”

And in the discussion we now include the new paragraph:

Lines (320-332): “Though the model here resolves only one bulk type of AOO and NOO each with fixed parameters, the framework developed may be useful for linking patterns of

diversity among multiple types to biogeochemical patterns. Our model shows that DIN concentrations are sensitive to the assumptions of the hypothesized yield and affinity differences (Fig. 4e--g), and this sensitivity may be exploited to gain insight into the controls on nitrifier diversity. For example, observations show a shift in dominance from ammonia-oxidizing bacteria (AOB) to ammonia-oxidizing archaea (AOA), as well as to different clades of AOA, from coastal to more oligotrophic environments and with an increase in depth (cite). Since larger AOB have lower affinity for NH₄ than AOA (cite), AOB should have a larger R*, which may contribute to an explanation for the observed decrease in [NH₄]:[NO₂] across the productivity gradient of the California Current (cite). We note, however, that time-varying fluxes of NH₄ from faster growing organisms closer to the dynamic mixed layer make it less likely that the steady state approximation should hold for NH₄, and so [NH₄] may be less predictable than [NO₂]. Eddy circulation, not resolved in this global model, will also contribute spatial and temporal heterogeneity to the basin-scale patterns simulated here (cite).”

We also have added the statement (line 309) **“We expect a diversity of yields and affinities to characterize the marine nitrifying community.”** in the discussion of the yield and efficiency uncertainties to emphasize this as well, and have also expanded on the deep water column results, which now read (line 192): **“Assuming no other sinks for [NO₂], we hypothesize that deep [NO₂], concentrations still reflect the R*s of the NOO, but of a diverse community. Affinities and efficiencies may vary with clades, such as those with maximum growth rates too low for survival at the PNM, or may be plastic in nature. A slower-growing, efficient, deep NOO clade could deplete NO₂ to lower levels.”**

Lastly, the authors end by stating that this model may provide insight into the fate of fixed nitrogen in anoxic regions. I think either this needs more explanation on how the model could do that or should be removed because I think it opens up an entirely separate complex discussion.

We agree that the previous statement was vague and have added more explanation: (line 361) **“The model can also be extended to examine the interactions of anaerobic as well as aerobic N-cycling metabolic functional types within an aerobic-anaerobic microbial ecosystem, which may provide insight into the fate of fixed nitrogen in anoxic zones.”**

We would like to leave this statement in because it will broaden the interested audience; for instance, Reviewer #3 noted an interest in this application. We believe that with the description of the general approach for such functional types within this manuscript (see lines 622-633 in Methods) and a further description of the generalized version including anaerobic functional types described in the Supplemental material (Supplementary Note 7) that it is worth retaining this application in the closing notes.

I think this paper deserves publication, and I applaud the authors for tackling a complex system and constructing a very sophisticated model that will be useful for numerous future studies.

We thank the reviewer for these positive comments.

Specific Comments:

Line 34: What does “each” refer to? Be more specific.

We have cut out this phrase for clarity and word limit.

Line 57: Reference?

This is our own speculation. We have moved this point to the discussion where the fact that it is a speculation is more clear and appropriate. (see **lines 295-297**)

Line 62: “rooted in the metabolisms of the two distinct clades nitrifying microorganisms”, what do you mean by this statement? Can you expand on this?

We have reworded this sentence as: **Lines 51-52: “Here, we will suggest that this reflects differences in the metabolisms of the distinct clades of nitrifying microorganisms.”**

Line 64: Here you mention subpolar regions as areas where [NO₂-] is elevated in the surface, but it also accumulates at HNLC’s (mentioned later), should you put both areas in this heading?

Thanks, yes, we have now changed to “**(line 53): are elevated in some areas, such as in subpolar regimes**” since we have not presented observations of HNLC regions in particular.

Lines 118-119: Confusing sentence, please expand.

Here we had meant that the three-fold difference directly reflects the half-reactions of NH₄ and NO₂-oxidation. However we decided to remove this sentence entirely because we originally thought it would clarify, not confuse, and the point is also made in a clearer manner in the methods, where half-reactions are written.

Line 121: Here is where you introduce uptake affinity, maybe this should be presented in the earlier section, “Microbial population dynamics.” In that section, you could spend more time explaining yield and affinity, since they come up many times in the discussion.

Thank you. Yes, we have added “**The uptake parameters together give an expression for the specific affinity (V_{max}/K), a measure of competitive strength at low resource concentrations (Fiksen et al 2013)**” into this earlier section (**lines 82-83**).

Line 123: What do you mean by stoichiometric and kinetic factors?

We have changed this to now read “**differences in yield and affinity.**”

Line 126: How well do we know that NOO are larger than AOO? How large do these differences need to be to make an affect on the affinity?

All known NOO are larger than the AOA (but not AOB) (to our knowledge), but it is possible that a smaller NOO may not yet have been discovered. Thus the size difference is a hypothesis. That this is a hypothesis was not made clear in the original text. We are careful in this revised version to add qualifiers “may be” and “if” instead of “is”. **Lines 107-108** now read: “**Second, we speculate that NOO specific uptake affinity may be lower than that of AOO, if (as is observed) NO₂-oxidizing bacteria are larger than NH₄-oxidizing archaea [cite Martens-Habben2009, Spieck2014].**”

We have also included detail about the relationship between size and specific affinity (“**specific affinity decreases with cell radius as r^2** ”), and the quantification of the effect of a 10-fold volume

difference: have added that this relates to “a 4.6-fold lower specific affinity” (lines 109-110).

Line 141: Could there be other limitations?

Certainly there can be other limitations, such as iron or other micronutrients. We are neglecting limitations by such micronutrients in the 1-D model, though we had done preliminary model experiments with an iron requirement and found that it made no difference to the solutions (see below). Also, when oxygen and DIN are both abundant, the growth rate is limited by internal constraints, which in the model is represented by the maximum uptake rate. Have added (line 128) “or by internal constraints (i.e., the maximum rate). For simplicity we neglect other limitations such as iron availability (Supplementary Note 3).”

Including this latter note warrants further explanation of the potential for iron limitation of the nitrifiers when discussing HNLC regions in the Discussion:

(line 280-282): “Additional model experiments that include an iron limitation to AOO and NOO growth show nearly identical solutions (Supplementary Note 3), perhaps reflecting that the relatively low nitrifier biomasses do not require much total iron.”

We thus add Supplementary Note 3 that briefly explains and discusses some simple model experiments (which were previously not mentioned) in which an iron limitation is introduced to nitrifier metabolism, with no significant effect on the results.

Line 149: I’m surprised that temperature does not affect growth rate of nitrifiers. Do you think this reference is applicable to all nitrifiers?

Probably not, and theoretically unlikely. But that was the only study we found that directly measured the temperature effect on marine nitrifiers, and we didn’t want to ignore its findings. Including temperature effect gives only a minor quantitative impact on the 1D results. We have included a new note in which we discuss the implications of these assumptions:

(Supplementary Note 1) : “In the illustrated simulations, temperature does not modify V_{max} , the maximum uptake rate of DIN by AOO and NOO that sets the maximum nitrification rate, in line with experimental evidence (cite Horak2013). Including a temperature modification changes solutions quantitatively but not qualitatively, with the exception of the oxygenated equatorial regions in the Indian and Atlantic basins in the global simulation. There, the solutions do exhibit a significant temperature sensitivity: If nitrifier rates are allowed to increase with temperature in the same way as the heterotrophic bacteria, the effect of the very high NO₂ concentrations (illustrated in Fig. 5 and discussed in the main text) is removed, and the resulting NO₂ concentrations are more similar to those of the modeled equatorial Pacific as well as to the observations. This may suggest that the observed lack of temperature sensitivity (cite Horak2013) may not apply to all nitrifiers. The temperature dependencies in general have a small but non-negligible effect on ecosystem structure, slowing all microbial rates with depth and with latitude.”

Figure 3a: Why does the model predict much higher [NO₂-] than the samples collected from the N. Pacific?

If we include both the affinity and yield differences that might exist between the two types of nitrifiers, then [NO₂] is overestimated (though note that the values still fall within the uncertainty range. To clarify, we added to the point balance results (lines 118-119) “underestimating

[NH₄]:[NO₂]", and have edited to the water column results (lines 183-184) **With both the yield and affinity differences between the nitrifiers included, [NH₄]:[NO₂] is about 1:10, and [NO₂] is overestimated."**

The sensitivity experiments shown in Fig 3e-g also emphasize this point. We make this clearer in the text too by rewording that paragraph (lines 181-189): **"We further demonstrate the control of [NH₄]:[NO₂], and its sensitivity to model assumptions, with model experiments in which we isolate the differences in yields and affinities of the NOO and AOO (Fig. 4e--g). With both the yield and affinity differences between the nitrifiers included, [NH₄]:[NO₂] is about 1:10, and [NO₂] is overestimated. When yields and affinities are assumed identical for AOO and NOO, using AOO parameters for both (giving them identical R*s), the profiles of [NH₄] and [NO₂] are identical, and [NO₂] is underestimated. With only a difference in yield *or* the uptake affinity, [NH₄]:[NO₂] is about 1:3 or slightly lower, as quantified with the point balance above, and consistent with observed [NO₂] and observed ratios (cite Santoro2013,Newell2013,Santoro2017). This suggests that either the yield or the affinity difference, or a smaller combination of both, may best represent natural assemblages of nitrifiers."**

Line 194-195: Another place that I noted needing more explanation of yield vs. affinity.

We have reworded this paragraph as cited just above (lines 181-189).

Lines 222-227: I found this paragraph confusing. I would consider rewriting it and highlighting the main point.

We have rewritten (lines 206-211):

Rates of NH₄ and NO₂ oxidation are identical below 150m depth for all model experiments (Fig. 4c,g). This matches the lack of consistent differences in observed rates (cite Ward2008, though differences in coastal waters have been documented (cite Heiss2016, Schaefer2017). Thus, differences in the modeled AOO and NOO emerge in nutrient distributions and nitrifier abundances, but not in subsurface nitrification rates. This reinforces our understanding that exported organic matter determines the rate of all steps in the sequence of remineralization metabolisms below the euphotic zone (cite Ward1988, Ward2008, Newell2013, Smith2016, Santoro2017).

Lines 238-240: I really like the idea of two PNM regimes.

Thank you, we do as well.

Line 251: I don't think you've referred to Figure 5 before you mention Figure 6c. Those figures should maybe be reordered.

Correct, we have reordered these figures.

Line 261: What do you mean by a higher R* organism? Is there a better descriptor you could use there?

Thanks for catching awkward wording: we meant an organism with a higher R*. However, we have now have removed this phrase from the end of the sentence to remain within the word limit.

Line 290: How did you choose 10uM? Ammonium and nitrite oxidation can occur at much lower oxygen concentrations (Bristow et al, 2016).

Yes, we do assume that these nanomolar oxygen concentrations occur within the secondary NO₂ maximum. Nitrification is able to still be active down to nanomolar concentrations as modeled with the diffusive oxygen limitation (Zakem and Follows 2016). However we do not want to get into such an analysis of this in this paper, but look only at the PNM (as stated in the introduction “**we focus here only on the dynamics of aerobic environments**”). We chose 10 uM as a value to completely remove any chance of nitrate reduction (we know there are some studies that have measured nitrate reduction at higher (20-30 uM O₂), and we have examined our resulting plot when the plotting threshold is 50um, instead of 10um, and it looks the same). We have rewritten and better explain this paragraph and plotting threshold:

(lines 258-260) “In Fig. 5, we illustrate the maximum [NO₂] in the water column where oxygen concentration is high (i.e., where O₂ >10 uM, which excludes the domain of anaerobic activity and the secondary NO₂ maximum).”

The updated caption for Fig. 5: “Only locations that co-occur with O₂ concentrations greater than 10 uM are plotted. (Note: (a) appears identical even with a plotting threshold of 50 uM.)”

Line 300: Could nitrite be transported out of low oxygen regions and sediments into the oxygenated waters, as another mechanism to reduction on particles? Also, is there a need to address nitrite production on particles throughout the article and how it affects the model results?

Transport of NO₂ out of the anoxic zones does occur in the global model, which does represent anaerobic activity and some NO₂ accumulation. Thus such transport is part of the results shown the model simulation. Our coarse simulation does however limit the resolution of some of the important physical processes in OMZs. Reduction within particles and transport out of sediments are processes not yet included in the model. Though interesting for future research, we have now cut this out in the revised paragraph because we don't experiment with or discuss particle dynamics elsewhere, and because it was quite a speculative point (i.e., we are not convinced this would make a difference to resulting NO₂ concentrations).

Lines 320-324: Can you add excretion of nitrite by phytoplankton into the model? Would it change the results?

We have explored adding phytoplankton excretion to the model and found that the results change quantitatively (the solutions show a higher accumulation of nitrite) but not qualitatively. However we decided not to include this process for the purposes of this study because (1) isotopic data do suggest that this is the main source of NO₂, and mechanism of accumulation are not obvious, and (2) on principle, we wanted to only include mechanisms that are explained more fundamentally, such as by energetics or physiology. We feel that a full exploration of this excretion of NO₂ as a consequence of energetic limitation, perhaps linking this to redox chemistry will be a fruitful future study.

Line 336: Can you add more on how ‘comammox’ would fit into the observations/model?

We have added **(lines 314-315): “measureable NO₂-oxidation rates may be lower if comammox metabolizes a portion of the NH₄ pool.”** Comammox has only been observed in biofilms, and the consensus, at least at this point in time, is that it is not expected in the

subsurface ocean, and so we would like to leave this speculation brief.

Lines 374-376: How much data are there backing the size differences between nitrite oxidizing bacteria and archaea? Are their ranges in sizes in organisms depending on location? What locations may this affinity difference not exist?

As discussed above, we have reworded our discussion to reflect that this is speculative and only relying on current understanding of NOO organisms. We have also extended the discussion and added a paragraph addressing patterns of abundances of small high-affinity AOA and larger AOB bacteria which should have a higher affinity mentioned:

(lines 319-331): Though the model here resolves only one bulk type of AOB and NOO each with fixed parameters, the framework developed may be useful for linking patterns of diversity among multiple types to biogeochemical patterns. Our model shows that DIN concentrations are sensitive to the assumptions of the hypothesized yield and affinity differences (Fig. 4e--g), and this sensitivity may be exploited to gain insight into the controls on nitrifier diversity. For example, observations show a shift in dominance from ammonia-oxidizing bacteria (AOB) to ammonia-oxidizing archaea (AOA), as well as to different clades of AOA, from coastal to more oligotrophic environments and with an increase in depth (cite). Since larger AOB have lower affinity for NH_4 than AOA (cite), AOB should have a larger R^* , which may contribute to an explanation for the observed decrease in $[\text{NH}_4]:[\text{NO}_2]$ across the productivity gradient of the California Current (cite). We note, however, that time-varying fluxes of NH_4 from faster growing organisms closer to the dynamic mixed layer make it less likely that the steady state approximation should hold for NH_4 , and so $[\text{NH}_4]$ may be less predictable than $[\text{NO}_2]$. Eddy circulation, not resolved in this global model, will also contribute spatial and temporal heterogeneity to the basin-scale patterns simulated here (cite)."

We think this is was very interesting and useful addition to the discussion and added additional words throughout the paper to emphasize that the R^* s should reflect the local community characteristics.

e.g. – lines 192-195: **"Assuming no other sinks for $[\text{NO}_2]$, we hypothesize that deep $[\text{NO}_2]$ concentrations still reflect the R^* s of the NOO, but of a more diverse community. Real nitrifying communities may have a distribution of affinities and efficiencies that may be plastic in nature, including some types with maximum growth rates too low for survival at the PNM. Lower grazing rates, higher efficiency, or higher affinity would all significantly lower R^* , allowing a slower-growing deep NOO clade to deplete NO_2 to lower levels."**

Line 415: What is the result of neglecting nitrous oxide formation?

We believe this is negligible, and have added this to the text with a citation: **(line 379-380) "which should have a negligible impact on AOB stoichiometry, given N_2O yields per mol N nitrified of less than 1% (cite Bange)."**

Line 437: Is this then assuming AOB and NOB have similar protein content? What if they are different?

Yes, we have assumed NOB nitrogen quota is similar to that of the AOB quota. We have now stated this explicitly **(line 406: "based on... a similar protein content for AOB and NOB")**. We highlight that an independent estimate of the N quota from the measurements of NOB of

Spieck et al. give a similar result, of about 1 fmol N per cell. We reworded this paragraph to emphasize the 10-fold difference in N quota and the uncertainty assumed for the quotas (0.07--0.16 and 0.7--1.6 fmol N cell for the two groups):

(lines 398-411): “Yields y_{NH_4} and y_{NO_2} were estimated from observations of cell growth on NH_4 or NO_2 , and the value of f was inferred for each using equations 6 and 7 (Supplementary Table~2). Some of the observed growth was mixotrophic, which exhibited yields about 10--20% higher than obligate chemoautotrophic growth (Supplementary Note 5). When required, the yield calculations assumed nitrogen cell quotas of 0.12 and 1.2 fmol N cell⁻¹ for the AOO and NOO groups, respectively, with a range of 0.07--0.16 and 0.7--1.6 fmol N cell⁻¹ contributing to uncertainty in the yields. These are computed from the 10.2+-1.1 fg protein cell⁻¹ content of ammonia-oxidizing archaea (AOA), as measured (cite{Martens-Habbena2009}), and an assumption of a 10-fold larger quota for the NOO, based on the measured minimum 10-fold difference in protein content between AOA and ammonia-oxidizing bacteria (AOB) (cite) and a similar protein content for AOB and NOB. The nitrogen content of protein was assumed to be 16% by weight, and uncertainty was incorporated by considering a range of 10% to 20% (Supplementary Table 2), giving the above nitrogen quotas. The NOO nitrogen quota can be independently estimated from the spheroidal volume of the new marine strain of *Nitrospina* of size 0.3-0.4 $\mu m \times 1-3 \mu m$ (cite): converting from the average bacterial carbon quota of 0.22 g C cm⁻³ with a C:N of 5 suggests a quota of order 1 fmol N cell⁻¹, consistent with our estimate.”

We assume that this large range of uncertainty may address the potential large differences between AOB and NOB, but also, the experiments that isolate/remove the effect of this difference in size also address this uncertainty more generally: our assumption of the size of NOB (though consistent with observations) results in just one of the many possible values of the NOB affinity.

Line 454: How representative is *Nitrosopumilis* from ref 41? Do you expect there to be major differences in the oceanic AOO.

We do not believe that there will be a dramatic difference between *Nitrosopumilis* and oceanic AOO. We add to the Methods addressing the application of ref 41, and comparison with kinetics measurements of natural marine assemblages (Peng et al 2016), which now reads as:

(lines 423-426): “Kinetics experiments with cultured NH_4 -oxidizing archaea *Nitrosopumilis* provide values for the parameters for the uptake of NH_4 by AOO (cite Martens-Habbena2009) (Supplementary Table 1), including the information needed to convert to a maximum specific uptake rate V_{max} (in mol NH_4 mol biomass N d⁻¹), and allows us to incorporate values that reflect a meaningful specific affinity for the AOO.

(lines 441-447): Measurement of the kinetics of natural assemblies of marine NH_4 oxidizers (cite Peng2016) show a lower half-saturation constant than *Nitrosopumilis* (27.2+-4.4 vs 133+-38 nM NH_4) with respect to a bulk NH_4 oxidation rate of 24.9+-1.3 nM N d⁻¹. We would not expect the half-saturation constants to be identical, because they should vary with the maximum rate, which is why the specific affinity (V_{max}/K) is the relevant trait (cite Fiksen2013). However, we can infer from this comparison that natural assemblages may have a lower maximum rate, and thus that the model here may overestimate nitrification rates in some locations.”

We believe however that our results would qualitatively be the same even with different parameters (see e.g. Fig 3 shaded area). The significant limitation in our study is single set of

AOO (and NOO) parameters as elaborated in the discussion section (**line 319**).

Line 509: Why were the nitrite oxidation rate incubations unsuccessful? Does that mean there were zero rates?

They were unsuccessful because incomplete reduction of the remaining *in situ* spiked NO₂ inhibited measurement of the enriched NO₃ pool. No, this does not mean that NO₂ oxidation rates were zero. We have expanded this discussion to

(lines 490-492): “A second treatment using 15NO₂ to measure the rate of NO₂ oxidation was unsuccessful because of the incomplete reduction of the remaining *in situ* spiked NO₂ inhibited measurement of the enriched NO₃ pool (which does not mean that NO₂ oxidation rates were zero).”

Reviewer #3 (Remarks to the Author):

Nitrate (NO_3) is the prevailing form of fixed inorganic nitrogen in the (interior of the) ocean. Nitrification is the microbiological process converting ammonium (NH_4), which is the inorganic nitrogen species released during organic matter degradation, to NO_3 . The intermediate product of this reaction, nitrite (NO_2), is rarely observed in the ocean, except for a thin layer at the bottom of the euphotic zone, the primary nitrite maximum (PNM) and for the mixed layer of (sub)polar waters. Generations of marine biologists and marine chemists had heard about the PNM at University, and certainly many have discussed the various theories to explain the PNM that have been put forward.

Zakem et al. use ocean models which apply rules of resource competition theory (for the first time) to tackle this problem. To start with, the authors provide a careful analysis of the stoichiometry and energetics of the two major agents of nitrification, ammonium oxidising organisms (AOO) and nitrite oxidising organisms (NOO). This careful analysis forms the basis of the developed models. Increasing the complexity from a 0D case (their point balance solution), through a vertical 1D model to a global (3D) ocean model they carefully introduce the elements of resource competition theory needed for their case and the interactions between biological processes and physical ones (mixing).

The major findings are that the surface ocean's lack of nitrite in oligotrophic euphotic zones is a consequence of competitive exclusion of nitrifying organisms by phytoplankton (which can make use of all forms of inorganic nitrogen) and that, in turn, limitation of phytoplankton by either light or iron, e.g. at the base of the euphotic zone and in high latitudes, allows for co-existence of slow growing chemoautotrophic nitrifiers and phytoplankton. Energetics and stoichiometry of the two steps of nitrification explain details of the vertical zonation (PNM, NH_4 -maximum).

The paper is well structured and well written. I also liked the details provided in the methods and supplementary information, putting the reader into the position to judge the uncertainties of parameters and hence the model results. The authors are also careful to stress limitations of their modelling approach, i.e. that it can't rule out by itself processes which have not been modelled. The authors follow a state-of-the-art open access policy and provide the 1D model to reviewers and readers. This contributes not only to reproducibility of science but may also, after publication, stimulate readers to adopt elements of the approach of Zakem et al.

The application of resource competition theory to the problem of global nitrite distribution is novel, in particular in this explicit modelling strategy. The argumentation is clear and benefits from the strategy of increasing the complexity, see above. Further the work, in particular its theoretical approach, the methodological developments and the availability of the code provide for a large likelihood that this publication will see follow up work, by the authors and others. One example is already obvious from the methods section, where the application of a variant of this model to questions of nitrogen loss in oxygen minimum zone (OMZ) somehow pops up. I see a very good potential that the application of resource competition theory, similar to what is demonstrated in this manuscript for nitrification, can help to disentangle the issue of anammox vs. heterotrophic denitrification (and all the other nitrogen conversions) in such waters.

After reading the manuscript for the first time I was unsure whether this work is topical enough, or interesting enough for researchers in related disciplines. Something necessary for a Nature journal, I guess. However, thinking about the potential of the approach described in this manuscript to improve also our understanding of OMZ nitrogen conversions, I am convinced the paper is interesting to many readers of Nature journals. And finally, this study will find many

readers, i.e. those that thought about the hows and whys of the PNM many year ago, when they entered the field of ocean research.

I suggest acceptance of the ms by Nature Communications with minor/technical revisions (s. below).

Thank you for these nice words, and such a good summary of our work.

A have a very few minor remarks:

Caption Fig. 3, panel c. ‘Observations of [NO₂-] and NH₄⁺ oxidations rates ...’, I guess it shall be ‘..of NO₂- and NH₄⁺ oxidations rates ...’, right?

Thanks, we realized this was not clear. (We had meant that only NO₂ concentration in panel a. and NH₄ oxid. rate in panel c. had these open circles.) We have altered the caption to now read **“Observations below the detection limit are indicated with open (vs. filled) markers.”**

Lines 242-244: The initial sentences introducing the global model were a little irritating. I first wandered why you now move from 1D to 3D, why don’t you simulate subpolar system with a 1D model here. I am not saying you should, but perhaps you could introduce the need for the 3D model not just with the subpolar problem.

Good point. We have changed to motivate understanding the global distribution, rather than just the subpolar problem, as **(lines 225-227): “Why does NO₂ accumulate at the subsurface in the subtropics, but also throughout the surface in subpolar regions to similar magnitudes in Fig. 1? To understand this distribution...”**

Lines 289-302. This is the only paragraph which I found somewhat difficult to read, since the argumentation is a little winding. I suggest to leave out the 2nd sentence (290-293), then write: ‘Both models ... (Fig. 6). However, this ...’. Finally, I was irritated by ‘The discrepancy could ... (301-302), which discrepancy? At the beginning of the paragraph you pointed out that observations and model agree. Fig. 6 caption: I suggest to mention also in the caption (like in the text) that is is the max of NO₂ for O₂>10.

Thank you, we have rearranged and reworded much of this paragraph, and added a note about the plotting threshold into the figure legend.

(lines 258-268): “In Fig. 5, we illustrate the maximum [NO₂] in the water column where oxygen concentration is high (i.e., where O₂ >10 μM, which excludes the domain of anaerobic activity and the secondary NO₂ maximum). The model captures the lowest values in the subtropical gyres, and higher values in high latitudes and in equatorial upwelling regions. The model does overestimate the values in the North Atlantic and Indian Equatorial regions. However, the coarse resolution of the physical model does not allow for the sharpness of equatorial circulation to be captured. And additional sensitivity studies show that these results can be sensitive to assumptions of temperature effects on heterotrophic bacteria (see Supplementary Note 1). The model also underestimates the highest values in the Pacific Equatorial region, although the climatology might be biased by aggregating the effects of eddies, showing high [O₂] and [NO₂] at the same location that were not measured at the same time. Mismatches between modeled and observed magnitudes could also indicate unaccounted-for diversity among the nitrifying community, which we discuss below.”

And in the caption for Fig. 5: “Only locations that co-occur with O₂ concentrations greater than 10 uM are plotted. (Note: (a) appears identical even with a plotting threshold of 50 uM.)”

Reviewer #3 Additional Comments:

Line 422: Could you provide a little more detail how you arrived at the equation $d=4c+h-2o-3n$. We realise that we had not explained c,h, etc. We have expanded with defining these symbols and an explanation of how the numbers come from biomass stoichiometry and redox state of inorganic constituents:

(lines 387-389) “Following previous methodology d represents the number of electron equivalents that correspond to the oxidation states of the inorganic constituents of that synthesis (cite). Assuming generic microbial biomass composition of C₅H₇O₂N and $d=4(5)+1(7)-2(2)-3(1)$ gives $d=20$.”

Line 468: References 74 and 75 after d^{-1} has to be given in different format.

Thanks, have put these references earlier in the sentence.

Line 491-492: Closing bracket of ‘(calculated ...’ is missing, I think.

Closed, thanks for catching that!

Line 585: Advection for 1D model is not clear, in particular since in the code it states ‘No advection’ Supporting Material.

There is vertical mixing in the 1D model, but the advection routine only solves for the sinking of organic matter. We add “for the sinking organic matter pool” (line 594) to make this clear. In the code itself, the “no advection” just was a note that there is no additional “w” to contribute, so “w” is set to zero before adding the sinking velocity “wd.” I have edited the comments in the code.

(General note: thank you so much for taking the time to find all of these inconsistencies!! Very much appreciated.)

Section S1.2 Several times you refer to some text ‘above’ (line 23, 28, 29). It seems that this suppl text has been extracted from a differently structured document. Please adjust such that the context is better linked to, e.g. by pointing to the main text instead of ‘above’.

Correct. This inconsistency came from a restructuring of an older version of the text. We have changed all, mostly to referencing Eqns. 4 and 5 in the main text.

Tab. S2 caption: Perhaps point out ‘in the 1D-water column model.’

Thank you; we have added this, and made it clearer that these parameters were for both the 3D and 1D models unless otherwise noted (and included a note that the physical parameters were only for the 1D model):

Supplementary Table 1 (pg. 7 of Supplement). caption: “Model parameters for 1D and 3D configurations, including the ranges of uncertainty from which values were randomly

sampled in the 1D water column model ensemble,” and subheading: “Physical parameters for 1D model.”

REVIEWERS' COMMENTS:

Reviewer #1 (Remarks to the Author):

The authors have done a nice job of addressing my comments in their revision. The role of grazer control in the calculation of R^* , and in determining nutrient concentrations, is much clearer.

Reviewer #2 (Remarks to the Author):

I applaud the authors in depth response to all of my comments in the first review. I am very pleased with the changes they have made, particularly in the discussion. After rereading the manuscript and supplemental material I have no further edits. I very much enjoyed reviewing this paper and I am pleased to recommend publication.